# Sampling Theory and Overparameterization: Shaping Loss Landscapes in $\ell^2$ Regression

## Abstract

Overparameterization in neural networks has demonstrated remarkable advantages for both memorization and generalization, particularly in models trained with gradient descent. While much of the existing research focuses on the interplay between overparameterization and gradient-based methods, we explore its influence on the loss landscape of $\ell^2$ supervised regression problems, independent of any specific optimizer. By leveraging the Nyquist-Shannon-Whittaker sampling theorem, we establish a theoretical link between sampling theory and overparameterized neural networks. Our findings reveal that overparameterization not only exponentially increases the number of global minima but also expands the dimensionality of loss valleys for various $\ell^2$ regression problems modelled with feedforward neural networks. We empirically validate these theoretical insights across multiple supervised $\ell^2$ regression tasks, trained with both gradient-based and non-gradient-based optimization algorithms. These results offer fresh perspectives on the advantages of overparameterization in neural network design, independent of the chosen learning algorithm.

## 1 Introduction

Overparameterization has demonstrated remarkable benefits for both memorization and generalization, particularly when training with gradient descent. Traditional learning theory suggests that models with excessive capacity are prone to overfitting. However, modern deep learning research has shown that overparameterized models can perfectly fit or memorize training data while still generalizing well to unseen data (Zhang et al., 2021). This memorization effect is particularly pronounced when using gradient descent, which efficiently navigates high-dimensional parameter spaces to locate global minima of the loss function, even in highly overparameterized networks (Arora et al., 2019; Zhang et al., 2021). The neural tangent kernel (NTK) theory has revealed that with sufficient overparameterization, gradient descent (flow) closely mirrors the behavior of kernel regression (Jacot et al., 2018; Bietti & Mairal, 2019; Huang et al., 2020). This insight highlights the critical role of overparameterization in understanding the dynamics of gradient descent in neural networks. Moreover, gradient descent exhibits an implicit bias toward finding solutions with minimal norm in overparameterized models, such as those with ReLU activations. This bias has been associated with improved generalization properties, even when the model can perfectly memorize the training data (Du et al., 2018; Allen-Zhu et al., 2019). This balance between memorization and generalization underscores the effectiveness of overparameterization in modern deep learning.

While these works reveal a deep connection between overparameterization, memorization, and generalization, they focus on the context of gradient descent as the learning algorithm. In this article, we seek to understand whether overparameterization offers inherent benefits for the loss landscape associated with $\ell^2$ supervised regression problems, independent of any particular optimizer.

Our approach builds on the sampling theory of Nyquist-Shannon-Whittaker (NSW) (Nyquist, 1928; Shannon, 1948; Whittaker, 1915) a foundational result in signal processing, providing the conditions under which a continuous signal can be perfectly reconstructed from a discrete set of samples. It states that if a signal is band-limited, meaning its frequency components are restricted to a maximum frequency $\omega_{max}$, then the signal can be fully recovered from its samples, provided the sampling rate is at least twice the highest frequency present—this rate is known as the Nyquist rate. Specifically, if the sampling interval $T$ satisfies $\frac{1}{T} \geq 2\omega_{max}$, the original signal can be reconstructed using a sum of

shifted sinc functions, where $\text{sinc}(x) = \frac{\sin(\pi x)}{\pi x}$ for $x \neq 0$ and $\text{sinc}(0) = 1$. This theorem is critical in modern data acquisition and reconstruction, ensuring that no information is lost in the sampling process as long as the Nyquist criterion is satisfied.

We build on these insights to establish a connection between sampling theory and supervised regression problems. Our first main result focuses on sinc-activated feedforward networks for modeling $\ell^2$ regression problems and demonstrates how overparameterization leads to an exponential increase in global minima around the origin of the parameter space. What is particularly interesting about this result is that it is independent of any optimizer, implying that for such networks, overparameterization provides a significant benefit for the loss landscape that should help any optimizer. Our second main result shows how sampling theory with the triangular function offers a new perspective on understanding ReLU feedforward networks. We mathematically prove that overparameterization results in an increase in the dimension of global minima that manifest as loss valleys in the parameter space.

Both theorems present a novel viewpoint on the benefits of overparameterization, going beyond what has been previously studied in the literature. To validate that our theoretical results provide practical insights into $\ell^2$ supervised regression problems, we conduct a series of experiments for both sinc and ReLU-activated feedforward networks using first-order gradient-based optimizers, second-order gradient-based optimizers, and non-gradient-based genetic optimizers. In each case, our results support our theoretical findings. We believe that the insights offered by sampling theory will lead to a deeper understanding of overparameterization and its effects on deep learning.

Our main contributions are:

1. Theoretical results explaining how overparameterization alters the global minima of the loss landscape for a supervised $\ell^2$ regression problem modelled with a sinc or ReLU-activated feedforward network that is independent of any optimizer.

2. A comprehensive validation of our theoretical results across a variety of supervised $\ell^2$ regression problems trained with different optimizers.

## 2 NOTATION

Within the course of this article we will use the following mathematical notations and definitions. The function sinc will be used throughout and is defined by $\text{sinc}(x) = \frac{\sin(\pi x)}{\pi x}$ for $x \neq 0$ and $\text{sinc}(0) = 1$. We will also make use of the Hilbert space $L^2(\mathbb{R})$, which we remind the reader is defined as the space of square integrable real valued functions on $\mathbb{R}$ with the Lebesgue measure, with inner product defined by $< f, g >_{L^2(\mathbb{R})} = \int_{\mathbb{R}} f \cdot g$. Given a point $z \in \mathbb{R}^n$ we will denote the open ball of radius $R$ about the point $z$ by $B_R(z)$. We will say two topological spaces $X$ and $Y$ are homeomorphic if there exists a continuous bijective function $\xi : X \to Y$ with a continuous inverse $\xi^{-1} : Y \to X$. The term closed interval will be used to mean an interval of the form $[a, b]$ which is defined as the set of real numbers $c \in \mathbb{R}$ that satisfy the inequality $a \leq c \leq b$ where $a, b \in \mathbb{R}$. We will primarily deal with feedforward networks as defined in standard texts such as Prince (2023). The parameter space for such a network will be denoted by $\mathbb{R}^{\text{param}}$ and will consist of all the weights and biases of the network. Finally, by the term overparameterization we mean that there are more parameters than data points. In general, we will often be considering situations where we add more neurons to the hidden layer of a shallow neural network and this is the primary way we will add extra parameters to our network. In the appendix we consider the case of deep networks where we increase parameters by adding hidden layers. For more details on notation see App. A.1.

## 3 RELATED WORK

Research on overparameterization has enhanced our understanding of how large models achieve both memorization and generalization. While overfitting was once a concern, studies like Zhang et al. (2021) showed that overparameterized networks can still generalize well, despite perfectly fitting training data. The neural tangent kernel (NTK) framework (Jacot et al., 2018) explained how gradient descent in these regimes resembles kernel regression, with Arora et al. (2019) and Huang et al. (2020) demonstrating that overparameterization smooths the loss landscape, leading

to multiple global minima. Additionally, Allen-Zhu et al. (2019) and Du et al. (2018) explored how gradient descent's bias toward minimal-norm solutions improves generalization of a network. Recent work by Belkin et al. (2019) and Nakkiran et al. (2021) introduced the "double descent" phenomenon, showing that increasing model size beyond the interpolation threshold further enhances performance. Despite this focus on gradient-based optimization, less attention has been given to overparameterization's impact on alternative optimizers.

The sinc function has been applied to neural networks in tasks such as audio sampling (Ravanelli & Bengio, 2018b;a) and dynamical systems (Ramasinghe et al., 2023; Saratchandran et al., 2024). Saratchandran et al. (2024) also established a universal approximation theorem for sinc-based networks. While these works focus on signal processing applications, this paper takes a different approach, using sinc networks to provide new insights into overparameterization.

# 4 OVERVIEW OF RESULTS

The problem we address in this paper is rooted in supervised $\ell^2$ regression. This machine learning task involves a dataset $\{(x_i, y_i)\}_{i=1}^n$, a neural model $\mathcal{N}(\theta; x)$, where $\theta \in \mathbb{R}^{\text{param}}$ represents the parameters and $x$ is the input variable, and an $\ell^2$ loss function $\mathcal{L}_2$, which is defined as:

$$\mathcal{L}_2(\theta) = \frac{1}{2n} \sum_{i=1}^n \left( \mathcal{N}(\theta; x_i) - y_i \right)^2. \tag{1}$$

The objective is to determine the parameters $\theta$ that minimize the loss function $\mathcal{L}_2$ through a suitable learning algorithm. While previous works have demonstrated the benefits of overparameterization when minimizing $\mathcal{L}_2$ using gradient-based algorithms, this paper seeks to understand whether overparameterization provides benefits for the loss function itself, independent of any specific optimizer.

Our approach is inspired by the classical Nyquist-Shannon-Whittaker (NSW) sampling theorem in signal processing (Martin, 1997). Sampling theory addresses the problem of reconstructing a signal $f$ from a collection of samples $\{f(x_i)\}_{i=1}^N$. Mathematically, if a function $f(t)$ is band-limited with a maximum frequency $\omega_{\max}$, the NSW theorem says that it can be reconstructed from its samples $\{f(nT)\}_{n \in \mathbb{Z}}$, provided the sampling rate satisfies $\frac{1}{T} \geq 2\omega_{\max}$ (known as the Nyquist rate). The reconstruction formula is given by:

$$f(x) = \sum_{n=-\infty}^{\infty} f(nT) \operatorname{sinc}\left( \frac{1}{T} (x - nT) \right). \tag{2}$$

In general, the theorem requires an infinite number of samples. As this is not possible in practice a finite but large $N > 0$ is usually chosen to produce the approximation

$$f(x) \approx \sum_{n=-N}^{N} f(nT) \operatorname{sinc}\left( \frac{1}{T} (x - nT) \right). \tag{3}$$

Fig. 1 gives a visual overview of the NSW sampling theorem. What is particularly striking about this theorem is that it provides an explicit formula for reconstructing a function based solely on discrete samples. In the case where one samples the signal at a sample rate less than the Nyquist frequencey signal cannot be accurately reconstructed and aliasing occurs (Martin, 1997). It is not difficult to see that the sum in equation 3 is a sinc activated shallow neural network.

We therefore see that we can reformulate the sampling problem as a supervised learning task. Given a band-limited signal $f$ and a dataset of samples $(nT, f(nT))_{n=-N}^N$, reconstructing $f$ can be viewed as minimizing the loss function $\mathcal{L}_2$ for a shallow sinc activated neural network $\mathcal{N}(\theta; x)$:

$$\mathcal{L}_2(\theta) = \frac{1}{2n} \sum_{i=1}^n \left( \mathcal{N}(\theta; nT) - f(nT) \right)^2 \text{ for } 1 \leq p < \infty. \tag{4}$$

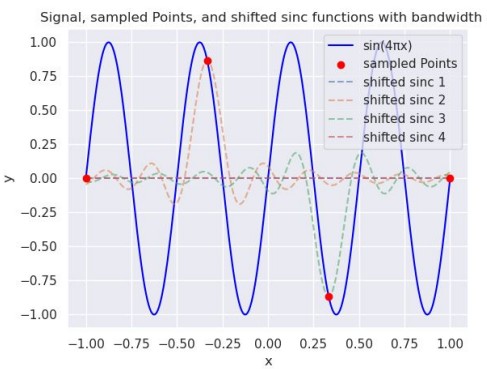 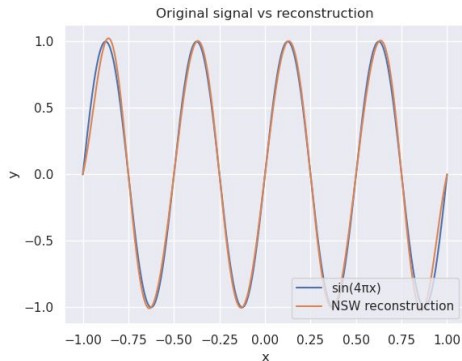

Figure 1: Left: Sampling the signal $sin(4\pi x)$ at $4$ different points and centering $\mathrm{sinc}$ functions with bandwidth $8$ about these points. Right: Using the NSW reconstruction formula 3 to reconstruct the signal from the sampled points.

This formulation establishes a clear connection between sampling theory and supervised learning. The goal of this work is to explore whether this perspective provides new insights into the benefits of overparameterization and its effect on the loss landscape of $\mathcal{L}_2$.

**Question:** Does the sampling theory approach of NSW for modelling signals lead to new insights for the benefits of overparameterization for networks modelling supervised $\ell^2$ regression problems?

Our results demonstrate that overparameterization has a substantial influence on the loss landscape of $\mathcal{L}_2$, independent of the specific optimization algorithm.

**Main results for sinc-activated networks:** For sinc-activated feedforward networks, our main theorem demonstrates that increasing the number of neurons, either by adding width or depth, leads to an exponential increase in the number of global minima for the loss function $\mathcal{L}_2$. These global minima are distributed around a ball centered at the origin of the parameter space. This result highlights a significant benefit for the loss landscape when the network becomes highly overparameterized, independent of the optimizer.

**Main results for ReLU activated networks:** For ReLU activated feedforward networks, we offer a novel perspective on overparameterization by framing it in terms of sampling with triangular functions. Our findings show that increasing the number of neurons, either by expanding the width or depth of the network, leads to a growth in the dimensionality of the global minima, which manifest as loss valleys. We provide a precise quantitative characterization of how this dimensionality increases. These results further highlight the significant benefits of overparameterization, regardless of the optimization algorithm used.

## 5 MAIN RESULTS

### 5.1 SINC ACTIVATED FEEDFORWARD NETWORKS

In this section, we present our main result on $\mathrm{sinc}$-activated neural networks for modeling $\ell^2$ supervised regression problems. To clarify the statement of the theorem, we first provide a precise definition of a loss valley that also constitutes a global minimum.

**Definition 5.1.** Let $\mathcal{L}_2 : \mathbb{R}^{\mathrm{param}} \to \mathbb{R}$ denote the $\ell^2$ loss function associated to a neural network as in equation 4. Let $\Lambda$ denote a collection of points in $\mathbb{R}^{\mathrm{param}}$ such that each $\theta \in \Lambda$ is a global minimum of $\mathcal{L}_2$. We say $\Lambda$ defines a global minimum valley if for each point $\theta \in \Lambda$ there exists an $r > 0$ such that $\Lambda \cap B_r(\theta)$ is homeomorphic to $\mathbb{R}^k$ for some $0 < k \leq \mathrm{param}$. The dimension of the global minimum valley is $k$. We say a point $\theta^*$ in $\mathbb{R}^{\mathrm{param}}$ is an isolated global minimum of $\mathcal{L}_2$ if there exists an $r > 0$ such that $B_r(\theta^*)\backslash\{\theta^*\}$ does not contain any global minima of $\mathcal{L}_2$.

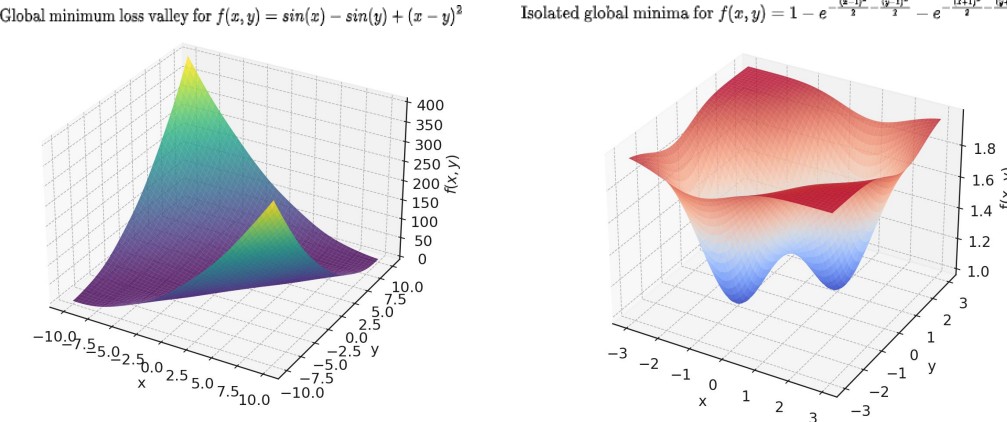

Figure 2: The function on the left admits a global minimum valley of dimension $1$ and the function on the right exhibits two isolated global minima.

In Fig. 2 we give examples of two different functions that exhibit a global minimum valley and isolated global minima.

**Theorem 5.2.** *Let $X = \{(x_i, y_i)\}_{i=1}^n$ be a labelled dataset of $n$ samples. Consider $\mathcal{N}(x; \theta)$ to be a shallow neural network with $n$ neurons in its hidden layer. Define the $\ell^2$ loss function $\mathcal{L}_2(\theta)$ as:*

$$\mathcal{L}_2(\theta) := \frac{1}{pn} \sum_{i=1}^n \left(\mathcal{N}(x_i; \theta) - y_i\right)^2, \tag{5}$$

*which is a mapping from the parameter space $\mathbb{R}^{param}$ to $\mathbb{R}$. Let $\mathcal{G}_R(n)$ denote the number of distinct global minimum valleys of $\mathcal{L}_2$ that intersect the ball $B_R(0)$ of radius $R \geq 1$, centered at the origin. The dependence on $n$ reflects the number of neurons in the hidden layer of the network $\mathcal{N}$.*

*If $l$ neurons are added to the hidden layer of $\mathcal{N}$, then the number of distinct global minimum valleys in $\mathcal{G}_R(n + l)$ grows at least exponentially in $l$.*

The proof of Thm. 5.2 can be found in App. A.1.1. Below, we provide an overview of the core ideas of the proof.

**Proof overview:** For this overview assume the data samples $(x_i)$ all lie in the interval $[0, 1]$ and are uniformly distributed. So let us assume $x_1 = \frac{1}{n}, x_2 = \frac{2}{n}, \ldots, x_{n-1} = \frac{n-1}{n}, x_n = 1$.

**Step 1:** The starting point is to think of the labelled dataset $X = \{(x_i, y_i)\}_{i=1}^n$ as defining a discrete signal $f$ with samples $(x_i)$ and sample values $(y_i) = (f(x_i))$. We then follow the approach of the NSW theorem equation 3, and centre a shifted $\mathrm{sinc}$ function with bandwidth $n$ about each $x_i$ having height $y_i$.

The sum defined by summing the shifted $\mathrm{sinc}$ functions

$$\sum_{i=1}^n y_i \mathrm{sinc}(nx - x_i) \tag{6}$$

then perfectly memorizes the data. This follows because $\mathrm{sinc}(m) = 0$ for any $m \in \mathbb{Z}\backslash\{0\}$. The next step is to observe that it can be implemented by a shallow $\mathrm{sinc}$ activated neural network $\mathcal{N}(x; \theta)$ where $\theta$ is defined as follows: The weight $W_1$ and bias $b_1$ in the first hidden layer are defined by

$$W_1 = [n, \ldots, n]^T \text{ and } b_1 = [-1, \ldots, -n]^T \tag{7}$$

and the weight $W_2$ and bias $b_2$ is taken to be

$$W_2 = [y_1, \ldots, y_n] \text{ and } b_2 = 0. \tag{8}$$

Then observe that $\mathcal{N}(x_i; \theta) = y_i$ showing that $\theta$ defined by equation 7, equation 8 is a global minimum of the loss function $\mathcal{L}_2$ defined in equation 5.

**Step 2:** Suppose we add one extra neuron to the hidden layer of $\mathcal{N}$. This then adds extra parameters to $\theta$ which we denote as

$$\widetilde{W}_1 = [W_1, a_1]^T \tag{9}$$

$$\widetilde{b}_1 = [b_1, a_2]^T \tag{10}$$

$$\widetilde{W}_2 = [W_2, a_3] \tag{11}$$

$$\widetilde{b}_2 = b. \tag{12}$$

Denoting all these parameters by $\widetilde{\theta}$ we find

$$\mathcal{N}(\widetilde{\theta}; x) = \sum_{i=1}^n y_i \operatorname{sinc}\left(nx - i\right) + a_3 \operatorname{sinc}(a_1 x - a_2) + b. \tag{13}$$

If we choose $a_1 = n$ and $a_2 \in \mathbb{Z} - \{1, \ldots, n\}$, $a_3 \in \mathbb{R}$ and $b = 0$. Then we see that any parameters $\theta^* = (\widetilde{W}_1, \widetilde{b}_1, \widetilde{W}_2, \widetilde{b}_2)$ satisfying these constraints yields

$$\mathcal{N}(\theta; x_i) = y_i \text{ for } 1 \le i \le n \tag{14}$$

which implies these $\theta^*$ parameterized by $\mathbb{Z} - \{1, \ldots, n\} \times \mathbb{R}$ are all global minimum valleys for the loss $\mathcal{L}_2$.

**Step 3:** The final step is to prove by induction that adding $l$ neurons leads to an increase in distinct global minimum valleys parameterized by $(\mathbb{Z}\backslash\{1, \ldots, n\})^l \times \mathbb{R}^l$. Thus to count how the number of distinct global minimum valleys within a ball $B_R(0)$ increase as $l$ gets bigger we need to understand how the set $B_R(0) \cap (\mathbb{Z}\backslash\{1, \ldots, n\})^l$ grows as $l$ gets bigger. For this we use a standard result that says that the number of integer points in a ball of radius $R \ge 1$, $B_R(0)$, about the origin in $\mathbb{R}^l$ grows exponentially with $l$, see Lem. A.3 in App. A.1.1. This completes the basic idea of the proof. $\qquad\square$

Thm. 5.2 also applies for deep $\operatorname{sinc}$ activated networks. The statement of the theorem in the deep case can be found in Thm. A.9 in App. A.1.2.

Thm. 5.2 establishes that overparameterization results in an exponential increase in global minimum valleys but it does not address whether these minima generalize well to points outside the training set. The following theorem addresses this by showing that for datasets obtained by sampling a signal $f \in L^2(\mathbb{R})$, many of the global minima given by Thm. 5.2 exhibit good generalization. The proof can be found in App. A.1.1.

**Theorem 5.3.** *Let $f \in L^2(\mathbb{R})$ be a continuous signal, and let $\epsilon > 0$ be a fixed threshold. Consider a dataset $(x_i, f(x_i))_{i=1}^n$ obtained by sampling $f$. Let $\mathcal{N}(\theta; x)$ be a shallow feedforward network with $\operatorname{sinc}$ activation and $n$ neurons in its hidden layer. Define the $\ell^2$ loss function based on the parameters $\theta$ of $\mathcal{N}$ as follows:*

$$\mathcal{L}_2(\theta) := \frac{1}{2n} \sum_{i=1}^n \left(\mathcal{N}(\theta; x_i) - f(x_i)\right)^2. \tag{15}$$

*If we add $l > 0$ neurons to the hidden layer of $\mathcal{N}$, for sufficiently large $l$, there are an infinite number of parameters $\theta$ lying in distinct global minimum valleys that satisfy the following bound:*

$$|f(x) - \mathcal{N}(\theta^*; x)| < \epsilon \tag{16}$$

*for any $x \in [0, 1] \setminus \{x_i\}_{i=1}^n$.*

## 5.2 ReLU ACTIVATED FEEDFORWARD NETWORKS

In this section we present our main result for ReLU activated networks. Our key insight is that a shallow ReLU network has the capacity to generate the triangle function with only 3 neurons in the hidden layer.

**Lemma 5.4.** *Let $T$ denote the triangle function defined by $T(x) = \max(1 - |x|, 0)$. Then*

$$T(x) = \operatorname{ReLU}(x + 1) + \operatorname{ReLU}(x - 1) - 2\operatorname{ReLU}(x). \tag{17}$$

*Furthermore, there exists a shallow ReLU neural network $\mathcal{N}$ with 3 neurons and a parameter $\theta^*$ such that $\mathcal{N}(x; \theta^*) = T(x)$. More generally, there exists a neural network $\mathcal{N}$ with 3 neurons and a parameter $\theta^*$ such that $\mathcal{N}(x; \theta^*) = T(\omega(x - a))$ for any $\omega > 0$ and any $a \in \mathbb{R}$.*

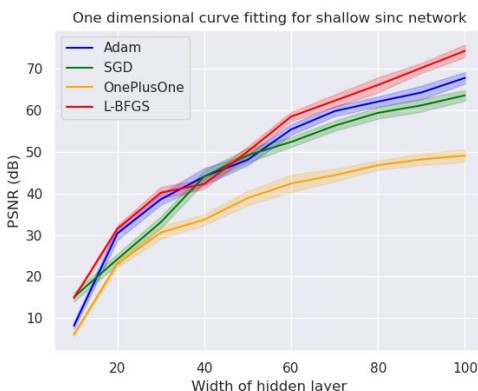 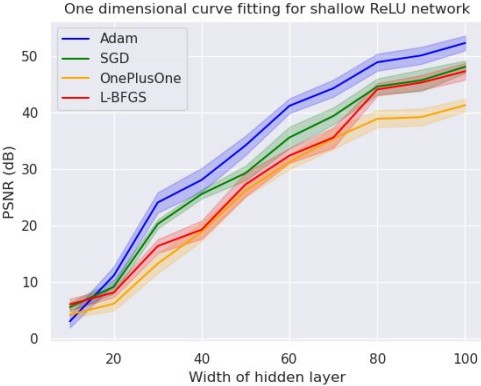

Figure 3: Final train PSNR after convergence is plotted against the width of the hidden layer for shallow $\mathrm{sinc}$ (left) and $\mathrm{ReLU}$ (right) networks, each trained with four different optimizers on a curve fitting task. The results show that, for both network types, increasing the width of the hidden layer consistently leads to higher train PSNR across all optimizers.

Using Lemma 5.4, we observe that a $\mathrm{ReLU}$ network can be interpreted as sampling with the triangular function. This key insight forms the foundation for the following theorem.

**Theorem 5.5.** *Let $X = \{(x_i, y_i)\}_{i=1}^n$ be a data set with $x_i \in \mathbb{R}^k$ and $y_i \in \mathbb{R}^m$. Let $\mathcal{N}(x; \theta)$ be a shallow $\mathrm{ReLU}$ neural network with $3n$ neurons in the hidden layer. Let*

$$\mathcal{L}_2(\theta) := \frac{1}{6n} \sum_{i=1}^{3n} (\mathcal{N}(x_i; \theta) - f(x_i))^2 \tag{18}$$

*denote the $\ell^2$ loss function. Then if we add $3l$ neurons to the hidden layer of $\mathcal{N}$, for $l > 0$, we have that there is an increase in global minima parameterized by the set*

$$\left( \mathbb{R} - \{ n \text{ closed intervals} \} \right)^l \times \left( \mathbb{R} - \{ n \text{ closed intervals} \} \right)^l \times \mathbb{R}^{ml}. \tag{19}$$

*In particular, we see that overparamterization leads to higher dimensional global minimum valleys whose dimension grows at worst linearly in $l$ i.e. the dimension grows as $\Omega(l)$.*

The proof of Thm. 5.5 proceeds analogously to the proof of Thm. 5.2 with the difference being that we centre a triangular function over each data point and then use Lem. 5.4. Details can be found in App. A.1.3. Furthermore, the results of Thm. 5.5 extends to deep $\mathrm{ReLU}$ networks. Details can be found in Thm. A.16 in App. A.1.4.

We also have a generalization theorem analogous to Thm. 5.3 for shallow $\mathrm{ReLU}$ feedforward networks. The reader can find the statement of this theorem and its proof in Thm. A.14 in App. A.1.3.

## 6 EXPERIMENTS

In this section, we aim to validate the results from Sec. 5.1 and 5.2. Thms. 5.2 and 5.5, along with their deep counter parts in App. A.1.2 and A.1.4, demonstrate that overparameterization facilitates the emergence of more global minima in the loss landscape, particularly near the origin of the parameter space. This implies that overparameterization should make it easier for an optimizer to find a global minimum.

To test our hypothesis, we conducted four common supervised $\ell^2$ regression experiments found in the literature: curve fitting, image regression, super resolution, and 3D shape modeling. For each task, we minimized the $\ell^2$ loss function $\mathcal{L}_2$ (see equation 5) using four distinct optimizers: SGD (a standard first-order method), Adam (an adaptive gradient-based optimizer), OnePlusOne (a gradient-free genetic algorithm), and L-BFGS (a second-order optimizer leveraging Hessian curvature). We

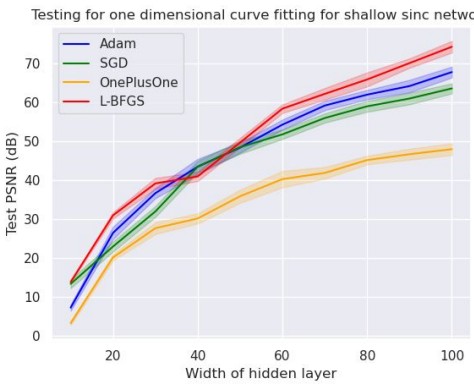 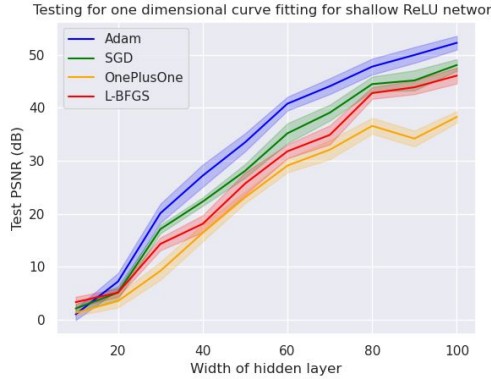

Figure 4: Final test PSNR after convergence is plotted against the width of the hidden layer for shallow $\mathrm{sinc}$ (left) and $\mathrm{ReLU}$ (right) networks, each trained with four different optimizers on a curve fitting task. The results show that, for both network types, increasing the width of the hidden layer consistently leads to higher test PSNR across all optimizers.

ran each experiment ten times, plotting the mean and standard deviation of the train PSNR after convergence across varying model widths and depths. Further experimental details are provided in appendix A.2.1. Consistently, we observed that overparameterization led to higher train PSNR values, indicating that it facilitated finding global minima for the optimizers.

## 6.1 CURVE FITTING

We consider the function $f(x) = \sin(2\pi x) + \sin(6\pi x)$ and use it to generate our dataset. Specifically, we select $x_i$ as 50 equally spaced points over the interval $[0, 1]$, with corresponding values $y_i = f(x_i)$. We then trained both shallow $\mathrm{sinc}$ and $\mathrm{ReLU}$ networks to regress the function $f$ using the $\mathcal{L}_2$ loss on a subset of 30 points out of the 50. As shown in Fig. 3, the PSNR increases as we add more neurons to the hidden layer, consistently improving across all optimizers.

We then obtained the test PSNR by testing on all the 50 points. As shown in Fig. 4, the test PSNR increases as we add more neurons to the hidden layer, consistently improving across all optimizers, validating the insight from Thm. 5.3 and Thm. A.14 in App. A.1.3.

## 6.2 IMAGE REGRESSION

In this experiment, our goal was to regress an image from the Div2k dataset. Given pixel coordinates $\mathbf{x} \in \mathbb{R}^2$, the task was to use a network $\mathcal{N}$ to predict the corresponding RGB values $\mathbf{c} \in \mathbb{R}^3$. Following the approach of Sitzmann et al. (2020), the dataset consisted of pixel coordinates paired with their respective RGB values. We trained $\mathrm{sinc}$ and $\mathrm{ReLU}$ deep networks of varying depths, ranging from 1 to 8 hidden layers, each containing 256 neurons, and employed the $\mathcal{L}_2$ loss, commonly used in image regression tasks (Sitzmann et al., 2020; Saratchandran et al., 2023; Saragadam et al., 2023). The results, shown in Fig. 5, demonstrate that increasing network depth consistently leads to higher PSNR values. However, we observed diminishing returns in PSNR improvement beyond 4 hidden layers, with the most significant gains occurring between 1 and 4 layers.

## 6.3 IMAGE SUPER RESOLUTION

In this experiment, we tackle an image super-resolution task. Following the methodology of Saragadam et al. (2023), we performed $4\times$ super-resolution on the Butterfly image from the DIV2K dataset. The problem is framed as solving $y = Ax$, where the operator $A$ applies $4\times$ downsampling. The goal is to recover $x$ using a feedforward network, with the task learned via the $\ell^2$ loss $\mathcal{L}_2$ as described in equation 5, similar to the approach in Saragadam et al. (2023). To enable testing (see App. A.2.2), we sampled $70\%$ of the total pixels in the image.

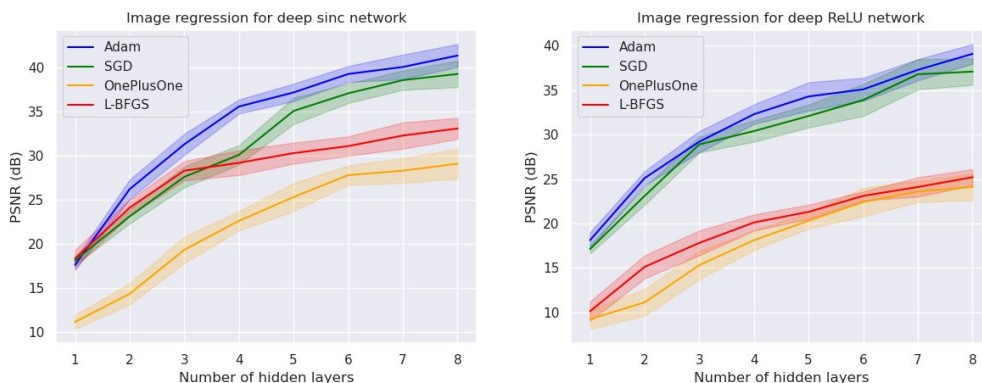

Figure 5: Final train PSNR after convergence is plotted against the number of hidden layers for deep sinc (left) and ReLU (right) networks, each trained with four different optimizers on an image regression task. The results show that, for both network types, increasing the depth of the network consistently leads to higher train PSNR across all optimizers.

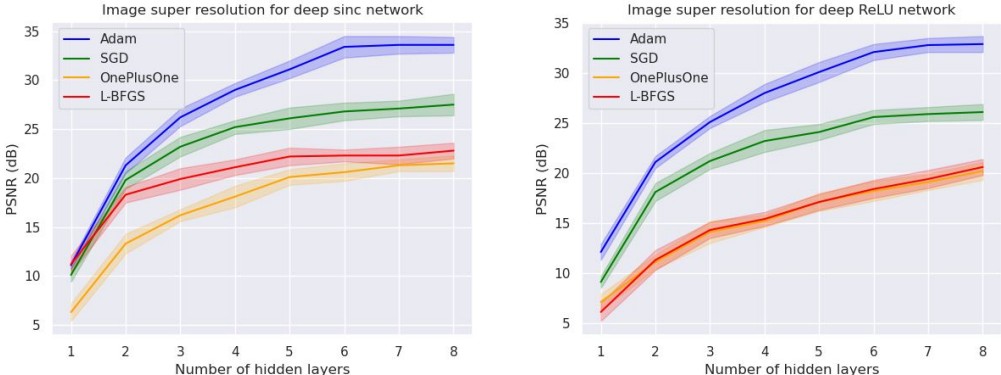

Figure 6: Final train PSNR after convergence is plotted against the number of hidden layers for deep sinc (left) and ReLU (right) networks, each trained with four different optimizers on an image super resolution task with training set consisting of 70% of the total pixels. The results show that, for both network types, increasing the depth of the network consistently leads to higher train PSNR across all optimizers.

We conducted the experiment using both sinc and ReLU-activated feedforward networks. The networks varied in depth, ranging from 1 to 8 hidden layers, each containing 256 neurons, and were trained using the $\mathcal{L}_2$ loss. The training results, presented in Fig. 6, demonstrate that increasing the network depth leads to higher PSNR values, although the improvements diminish beyond 4 hidden layers.

### 6.4 3D SHAPE MODELLING

In this experiment we optimize a binary occupancy field, which represents a 3D shape as the decision boundary of a neural network as in Wang et al. (2021); Gropp et al. (2020). We use the *Thai statue* instance obtained from the Stanford 3D Scanning repository. We trained sinc and ReLU deep networks of varying depths, each with 128 neurons, utilizing the loss $\mathcal{L}_2$ to regress the Thai statue. The results, shown in Fig. 5, indicate that increasing the depth of the networks consistently leads to higher PSNR values.

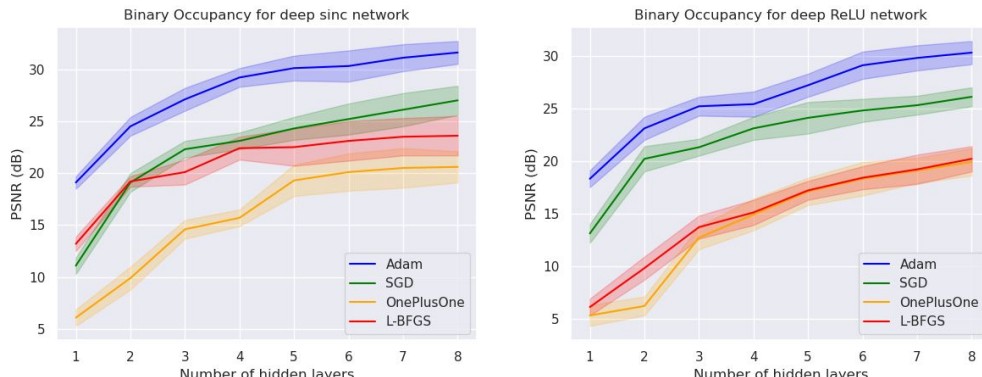

Figure 7: Final train PSNR after convergence is plotted against the number of hidden layers for deep sinc (left) and ReLU (right) networks, each trained with four different optimizers on a binary occupancy shape fitting task. The results show that, for both network types, increasing the depth of the network consistently leads to higher train PSNR across all optimizers.

## 6.5 FURTHER EXPERIMENTS

**Epochs needed for convergence:** Results on the number of epochs needed for each optimizer to converge as width and depth are added are carried out in App. 6.5.

**Testing for image super resolution:** Although our Thm. 5.3 and Thm. A.14 in App. A.1.3 focus on points outside the training set for one dimensional signals. We decided to empirically see what happens when we consider testing for higher dimensional signals. The results for super image resolution, following Sec. 6.3, are given in App. A.2.2.

**Testing for binary occupancy:** Results on testing for the binary occupancy experiment carried out in Sec. 6.4 are given in App. A.2.2.

**Neural Radiance Fields (NeRF):** We also carried out experiments on Neural Radiance Fields (Mildenhall et al., 2021). Results can be found in App. A.2.2.

## 7 LIMITATIONS

Our results in Theorems 5.3 and A.14 apply to signals in $L^2(\mathbb{R})$, as they are rooted in the Nyquist-Shannon-Whittaker sampling theorem, which pertains to such signals. An interesting extension would be to explore whether bounds outside the training data can be established for higher-dimensional signals in $L^2(\mathbb{R}^k)$ for $k > 1$. We believe this direction could be linked to the multidimensional sampling theorem by Petersen & Middleton (1962), potentially offering new insights into the role of network depth and its impact on generalization. We aim to take this up in a future project.

## 8 CONCLUSION

In this paper, we demonstrated that overparameterization, viewed through the lens of sampling theory, provides valuable insights into the structure of the loss landscape for $\ell^2$ supervised regression problems. Our theoretical findings reveal that both sinc and ReLU activated feedforward networks, when overparameterized, significantly increase the number of global minima for the $\ell^2$ loss function, regardless of the optimizer used. Empirical validation with various optimizers reinforces these results, highlighting the pivotal role of overparameterization. We hope these insights inspire new approaches to understanding neural networks and the loss functions used to train them.

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

# A  APPENDIX

## A.1  THEORETICAL RESULTS

In this section we give detailed proofs of the theorems from the main paper. For the sake of self containment we outline the theoretical notation we will be using throught this section.

**Theoretical Notation:**  We outline the main notation we will be using throughout this section. We remind the reader that the function $\text{sinc}$ is defined by $\text{sinc}(x) = \frac{sin(\pi x)}{\pi x}$ for $x \neq 0$ and $sinc(0) = 0$. We will also use the triangular function which is defined by $\text{T}(x) = \max(1 - |x|, 0)$. We will use the standard notation of $\mathbb{R}$ and $\mathbb{Z}$ to denote the real numbers and integers respectively. Closed intervals will be defined by the notation $[a, b]$ where $a, b \in \mathbb{R}$ with $a < b$, consisting of numbers $c \in \mathbb{R}$ such that $a \leq c \leq b$. The notation $(a, b)$ will denote an open interval of real numbers, where $a, b \in \mathbb{R}$ with $a < b$, consisting of numbers $c \in \mathbb{R}$ such that $a < c < b$. Open balls about a point $z \in \mathbb{R}^n$ will be denoted by $B_R(z)$ for any $R > 0$. We will say to topological spaces $X$ and $Y$ are homeomorphic if there exists a continuous function $f : X \to Y$ that is bijective and has a continuous inverse $f^{-1} : Y \to X$. The space $L^2(\mathbb{R})$ denotes the Hilbert space of square integrable real values functions on $\mathbb{R}$ with the Lebesgue measure, the inner product being defined by $< f, g >:= \int_{\mathbb{R}} f \cdot g$ for $f, g \in L^2(\mathbb{R})$.

All the neural networks we consider will be feedforward, as defined in Prince (2023), and denoted by $\mathcal{N}$. The parameter space of $\mathcal{N}$ will be denoted by $\mathbb{R}^{\text{param}}$. The objective functions used to train such networks will be the standard $\ell^2$ loss functions (Prince, 2023) for supervised learning tasks, which given a dataset $\{(x_i, y_i)\}$ and a feedforward network $\mathcal{N}(\theta; x)$, where $\theta$ are the parameters of the network, is defined by

$$\mathcal{L}_2(\theta) = \frac{1}{2n} \sum_{i=1}^{n} (\mathcal{N}(\theta; x) - y_i)^2. \tag{20}$$

An important observation that we will use through the paper is that if the following equations are satisfied for a parameter $\theta$

$$\mathcal{N}(\theta; x_i) = y_i \text{ for } 1 \leq i \leq n \tag{21}$$

then the parameter $\theta$ is necessarily a global minimum for $\mathcal{L}_2$. This follows from noting that by equation 20, we must have that $\mathcal{L}_2(\theta) \geq 0$ for any $\theta \in \mathbb{R}^{\text{param}}$.

### A.1.1  RESULTS FOR SHALLOW SINC NETWORKS

In this section we would like to give the proof of Thms. 5.2 and 5.3. In order to do this we will start with some propositions and lemmas.

**Proposition A.1.** *Let $X = \{(x_i, y_i)\}_{i=1}^{n}$ be a data set with $x_i \in \mathbb{R}^k$ and the labels $y_i \in \mathbb{R}^m$. Let $\mathcal{N}(x; \theta)$ be a* sinc *activated shallow neural network with $n$ neurons. Then there exists a parameter $\theta^*$ such that*

$$\mathcal{N}(x_i; \theta^*) = y_i \text{ for all } 1 \leq i \leq n. \tag{22}$$

*In particular, $\theta^*$ is a global minimum for the $\ell^2$ loss objective*

$$\mathcal{L}_2(\theta) := \frac{1}{2n} \sum_{i=1}^{n} (\mathcal{N}(x_i; \theta) - y_i)^2. \tag{23}$$

*Furthermore, we can write down an explicit expression for the parameter $\theta^*$.*

*Proof.* The main idea of the proof is that we can perform a reconstruction by centering suitable sinc functions around the domain data $x_i$ with height given by the labels $y_i$. Then one uses the insight that such a construction can be done via a shallow neural network with a sinc activation.

To begin with we will start by assuming the data set is one-dimensional so that $x_i \in \mathbb{R}$ and $y_i \in \mathbb{R}$. Let us normalize the data points so that $x_i = \frac{\widetilde{p}_i}{q_i}$ for $p_i, q_i \in \mathbb{Z}$. Then put each each $x_i$ over a common denominator $q$ and write $x_i = \frac{p_i}{q}$. Note that the fact we can do this is an assumption though one that is satisfied in practise due to the finite precision of the computers. This normalization can also be

interpreted from the signal processing viewpoint as assuming the bandwidth of the discrete signal defined by the data $\{x_i, y_i\}$ is a multiple of $q > 0$.

We then define a parameter $\theta^*$ for a sinc activated shallow neural network $\mathcal{N}$, with $n$ neurons in the hidden layer, as follows: The weight $W_1$ and bias $b_1$ of the hidden layer will be

$$W_1 = [q, \ldots, q]^T \text{ and } b_1 = [-p_1, \ldots, -p_n] \tag{24}$$

and the weight $W_2$ and bias $b_2$ of the output layer will be

$$W_2 = [y_1, \ldots, y_n] \text{ and } b_2 = 0. \tag{25}$$

If we input this parameter into $\mathcal{N}$ we obtain

$$\mathcal{N}(x; \theta^*) = y_1 \text{sinc}(qx - p_1) + \cdots + y_n \text{sinc}(qx - p_n). \tag{26}$$

Using the fact that $\text{sinc}(m) = 0$ for any integer $m \in \mathbb{Z} - \{0\}$ we find that

$$\mathcal{N}(x_i; \theta^*) = y_i \text{ for all } 1 \leq i \leq n. \tag{27}$$

This shows that $\theta^*$ is an explicit global minimum for the loss function $\mathcal{L}_2$.

In the case of higher dimensional data satisfying $x_i \in \mathbb{R}^k$ and $y_i \in \mathbb{R}$ we proceed as follows. We write the data as follows:

$$x_1 = [x_{11}, \ldots, x_{k1}]^T$$
$$\vdots$$
$$x_n = [x_{1n}, \ldots, x_{kn}]^T.$$

As in the one dimensional case, we normalize each data coordinate over a common denominator so that $x_{ij} = \frac{p_{ij}}{q}$.

Then we define a parameter $\theta^*$ for a shallow sinc activated neural network $\mathcal{N}$ as follows: The weight $W_1$ and bias $b_1$ of the hidden layer will be:

$$W_1 = \begin{bmatrix} q & \cdots & q \\ \vdots & \vdots & \vdots \\ q & \cdots & q \end{bmatrix} \text{ and } b_1 = \begin{bmatrix} -(p_{11}+ & \cdots & +p_{k1}) \\ \vdots & \vdots & \vdots \\ -(p_{1n}+ & \cdots & +p_{kn}) \end{bmatrix} \tag{28}$$

Note that in this case $W_1$ has dimensions $n \times k$ and $b_1$ $n \times 1$.

The weight $W_2$ and bias $b_2$ of the output layer will be

$$W_2 = [y_1, \ldots, y_n] \text{ and } b_2 = 0. \tag{29}$$

Then given an arbitrary input $z = [z_1, \ldots, z_k]^T$ we have

$$\mathcal{N}(z; \theta^*) = y_1 \text{sinc}\left( qz_1 - p_{11} + \cdots + qz_k - p_{k1} \right) + \cdots$$
$$+ y_n \text{sinc}\left( qz_1 - p_{1n} + \cdots + qz_k - p_{kn} \right).$$

We then observe that using the fact that $\text{sinc}(n) = 0$ for all integers $n \in \mathbb{Z} - \{0\}$ we have

$$\mathcal{N}(x_i; \theta^*) = 0 \text{ for all } 1 \leq i \leq n \tag{30}$$

showing that the explicit parameter $\theta^*$ is a global minimum of the loss function $\mathcal{L}_2$.

The final step is to consider the case when the labels are also high dimensional. In particular, assume $x_i \in \mathbb{R}^k$ and $y_i \in \mathbb{R}^m$. In this case we write $y_i = (y_{1i}, \ldots, y_{mi}$, for $1 \leq i \leq n$, where We will also use the same convention we used for the data points $x_i$ above.

In this case the weights and bias of the hidden layer will be the same as in equation 28. The weight $W_2$ and bias $b_2$ for the output layer will be defined by

$$W_2 = \begin{bmatrix} y_{11} & \cdots & y_{1n} \\ \vdots & \vdots & \vdots \\ y_{m1} & \cdots & y_{mn} \end{bmatrix} \text{ and } b_2 = [0, \cdots, 0]^T. \tag{31}$$

For an arbitrary input $z = [z_1, \ldots, z_k]^T$ we have

$$\mathcal{N}(z; \theta^*) = [\mathcal{N}_1(z; \theta^*), \ldots, \mathcal{N}_m(z; \theta^*)]^T \tag{32}$$

where

$$\mathcal{N}_j(z; \theta^*) = y_{j1}\text{sinc}\bigg(qz_1 - p_{11} + \cdots + qz_k - p_{k1}\bigg) + \cdots$$

$$+ y_{jn}\text{sinc}\bigg(qz_1 - p_{1n} + \cdots + qz_k - p_{kn}\bigg)$$

for $1 \le j \le m$.

It is clear from this that

$$\mathcal{N}(x_i; \theta^*) = y_i \text{ for all } 1 \le i \le n. \tag{33}$$

This gives an explicit representation of $\theta^*$ in this setting and shows that it is a global minimum for the loss function $\mathcal{L}_2$. □

The following proposition seeks to understand what happens if we add a single extra neuron to the above neural network found in Prop. A.1.

**Proposition A.2.** *Let $X = \{(x_i, y_i)\}_{i=1}^n$ be a data set with $x_i \in \mathbb{R}^k$ and labels $y_i \in \mathbb{R}^m$. Let $\mathcal{N}(x; \theta)$ be a* sinc *activated shallow neural network with $n$ neurons given by Prop. A.1. Let*

$$\mathcal{L}_2(\theta) := \frac{1}{2n} \sum_{i=1}^n (\mathcal{N}(x_i; \theta) - y_i)^2 \tag{34}$$

*denote the $\ell^2$ loss function.*

*Then adding $1$ extra neuron to the hidden layer of $\mathcal{N}(x; \theta)$ results in an increase in global minima of $\mathcal{L}_2$ parameterized by the set $\mathbb{Z} \times \mathbb{Z} - \{n \text{ points}\} \times \mathbb{R}^m$. Furthermore, we can write down explicit expressions for each of these new global minima.*

*Proof.* We start by assuming $x_i \in \mathbb{R}$ and $y_i \in \mathbb{R}$. We assume our data is normalized so that $x_i = \frac{p_i}{q}$ as in the proof of Prop. A.1.

The proof will proceed by building the extra global minima using the structure of $\theta^*$ found in Prop. A.1.

Since we have added one extra neuron to $\mathcal{N}$ the weight $W_1$ in the hidden layer will have dimension $(n+1) \times 1$ and the bias $b_1$ will have dimension $(n+1) \times 1$. The weight $W_2$ in the output layer will have dimension $1 \times (n+1)$ and the bias $b_2$ will have dimension $1 \times 1$.

Let us define the weight $W_1 = [q, \ldots, q, \lambda_1]^T$ and bias $b_1 = [-p_1, \ldots, -p_n, \eta]^T$ where for now $\lambda_1$ and $\eta$ are parameters in $\mathbb{R}$. Let the weight of the output layer be given by $W_2 = [y_1, \ldots, y_n, \lambda_2]$ where $\lambda_2 \in \mathbb{R}$ and the bias by $b_2 = 0$. If we input these parameters into the neural network $\mathcal{N}$ we obtain:

$$\mathcal{N}(x; \theta) = y_1\text{sinc}(qx - p_1) + \cdots + y_n\text{sinc}(qx - p_n) + \lambda_2\text{sinc}(\lambda_1 x - \eta).$$

We now want to show that for the right choices of $\lambda_1, \lambda_2, \eta \in \mathbb{R}$ we can obtain parameters $\theta^*$ so that $\mathcal{N}(x_i; \theta^*) = y_i$. In order to see what parameters we can choose, write $\lambda_1 = q\xi$ where $\xi \in \mathbb{Z}$ then let $\eta \in \mathbb{Z} - \{\xi p_1, \ldots, \xi p_1\}$ and let $\lambda_2 \in \mathbb{R}$.

Observe that for these parameters we have that

$$\lambda_2\text{sinc}(q\xi x_i - \eta) = 0 \text{ for } 1 \le i \le n \tag{35}$$

where we have used the fact that $\text{sinc}(m) = 0$ for any integer $m \in \mathbb{Z} - \{0\}$. We thus see that if we define $\theta^*$ by $W_1 = [q, \ldots, q, \xi q]^T$, $b_1 = [p_1, \ldots, p_n, \eta]$, $W_2 = [y_1, \ldots, y_n, \lambda]$ and $b_2 = 0$ where $\xi \in \mathbb{Z}, \eta \in \mathbb{Z} - \{\xi p_1, \ldots, \xi p_n\}$ and $\lambda \in \mathbb{R}$ then for such a parameter we have that

$$\mathcal{N}(x_i; \theta^*) = y_i \text{ for } 1 \le i \le n. \tag{36}$$

This proves the theorem for the setting of one dimensional data.

In the case that $x_i \in \mathbb{R}^k$ and $y_i \in \mathbb{R}$, we start by writing the data as follows:

$$x_1 = [x_{11}, \ldots, x_{k1}]^T$$
$$\vdots$$
$$x_n = [x_{1n}, \ldots, x_{kn}]^T.$$

As in the one dimensional case, we write each data coordinate over a common denominator so that $x_{ij} = \frac{p_{ij}}{q}$.

We then write the weight $W_1$ and bias $b_1$ of the hidden layer as

$$W_1 = \begin{bmatrix} q & \cdots & q \\ \vdots & \vdots & \vdots \\ q & \cdots & q \\ \xi_1 q & \cdots & \xi_k q \end{bmatrix} \text{ and } b_1 = \begin{bmatrix} -(p_{11}+ & \cdots & +p_{k1}) \\ \vdots & \vdots & \vdots \\ -(p_{1n}+ & \cdots & +p_{kn}) \\ -\eta \end{bmatrix} \tag{37}$$

where $\xi_1, \ldots, \xi_k \in \mathbb{R}$ and $\eta \in \mathbb{R}$. Note that in this case $W_1$ has dimensions $(n+1) \times k$ and $b_1$ $(n+1) \times 1$. We write the weight $W_2$ and the bias $b_2$ of the output layer as $W_2 = [y_1, \ldots, y_n, \lambda]$ and $b_2 = 0$ where $\lambda \in \mathbb{R}$.

If we insert these weights and biases into the neural network $\mathcal{N}$, for any input point $z = [z_1, \ldots, z_k]^T$ we obtain

$$\mathcal{N}(z; \theta) = y_1 \text{sinc}(qz_1 - p_{11} + \cdots + qz_k - p_{k1}) + \cdots$$
$$+ y_n \text{sinc}(qz_1 - p_{1n} + \cdots + qz_k - p_{kn})$$
$$+ \lambda \text{sinc}(\xi_1 qz_1 + \cdots + \xi_k qz_k - \eta).$$

We then denote by $\theta^*$ those parameters that satisfy $\xi, \ldots, \xi_k \in \mathbb{Z}$ and $\eta \in \mathbb{Z} - \{\xi_1 p_{11} + \cdots + \xi_k p_{k1}, \ldots, \xi_1 p_{1n} + \cdots + \xi_k p_{kn}\}$ and $\lambda \in \mathbb{R}$ and observe that

$$\mathcal{N}(x_i; \theta^*) = y_i \text{ for } 1 \le i \le n. \tag{38}$$

This proves the theorem for the case $x_i \in \mathbb{R}^k$ and $y_i \in \mathbb{R}$.

For the general case of data $x_i \in \mathbb{R}^k$ and $y_i \in \mathbb{R}^m$ with $k, m > 1$ we proceed similar to the above case. The weights $W_1$ and bias $b_1$ of the hidden layer will be given by

$$W_1 = \begin{bmatrix} q & \cdots & q \\ \vdots & \vdots & \vdots \\ q & \cdots & q \\ \xi_1 q & \cdots & \xi_k q \end{bmatrix} \text{ and } b_1 = \begin{bmatrix} -(p_{11}+ & \cdots & +p_{k1}) \\ \vdots & \vdots & \vdots \\ -(p_{1n}+ & \cdots & +p_{kn}) \\ -\eta \end{bmatrix} \tag{39}$$

and the weights $W_2$ and bias $b_2$ of the output layer will be given by

$$W_2 = \begin{bmatrix} y_{11} & \cdots & y_{1n} & \lambda_1 \\ \vdots & \vdots & \vdots & \vdots \\ y_{m1} & \cdots & y_{mn} & \lambda_m \end{bmatrix} \text{ and } b_2 = [0, \cdots, 0]^T. \tag{40}$$

For an arbitrary input $z = [z_1, \ldots, z_k]^T$ we have

$$\mathcal{N}(z; \theta^*) = [\mathcal{N}_1(z; \theta^*), \ldots, \mathcal{N}_m(z; \theta^*)]^T \tag{41}$$

where

$$\mathcal{N}_j(z; \theta^*) = y_{j1} \text{sinc}(qz_1 - p_{11} + \cdots + qz_k - p_{k1}) + \cdots$$
$$+ y_{jn} \text{sinc}(qz_1 - p_{1n} + \cdots + qz_k - p_{kn})$$
$$+ \lambda_j \text{sinc}(\xi_1 qz_1 + \cdots + \xi_k qz_k - \eta)$$

for $1 \le j \le m$. We then denote by $\theta^*$ those parameters that satisfy $\xi, \ldots, \xi_k \in \mathbb{Z}$ and $\eta \in \mathbb{Z} - \{\xi_1 p_{11} + \cdots + \xi_k p_{k1}, \ldots, \xi_1 p_{1n} + \cdots + \xi_k p_{kn}\}$, $\lambda_1, \ldots, \lambda_m \in \mathbb{R}$ and observe that for such parameters we have

$$\mathcal{N}(x_i; \theta^*) = y_i \text{ for } 1 \le i \le n. \tag{42}$$

This shows that the extra global minima are parameterized by the set

$$\mathbb{Z} \times \mathbb{Z} - \{\xi_1 p_{11} + \cdots + \xi_k p_{k1}, \ldots, \xi_1 p_{1n} + \cdots + \xi_k p_{kn}\} \times \mathbb{R}^m \tag{43}$$

$\square$

We will need the following lemma about how the integer points in $\mathbb{R}^n$ within a ball $B_R(0)$ about the origin for $R > 1$ grows as the dimension $n$ increases.

**Lemma A.3.** *Let $R \geq 1$ and let $B_R(0)$ denote the ball of radius $R$ about the origin in $\mathbb{R}^n$. Let $\Lambda(n) = \mathbb{Z}^n \cap B_R(0)$ denote the integer lattice points in $B_R(0)$ and $|\Lambda(n)|$ denote its cardinality. Then $|\Lambda(n)|$ grows exponentially in $n$.*

The proof of this lemma can be found in Chamizo (1998) and Fricker (2013).

Note that the above lemma also holds true if we remove a finite set of points in $\mathbb{Z}^n$.

We now have all the ingredients to give a proof of Thm. 5.2 from the main body of the paper.

*Proof of Thm. 5.2.* The proof uses the result of Prop. A.2 and Lem. A.3.

We saw from the proof of Prop. A.2 that if we add 1 neuron to $\mathcal{N}$ there are a number of new global minimum valleys that are parameterized by the set $\mathbb{Z} \times \mathbb{Z} - \{$ n points $\} \times \mathbb{R}^m$. Following that proof we see that if we add $l > 0$ neurons to $\mathcal{N}$ we will get a collection of new global minimum valleys parameterized by the set $\mathbb{Z}^l \times (\mathbb{Z} - \{$ n points $\})^l \times \mathbb{R}^{lm}$.

Observe that the connected components of $\mathbb{Z}^l \times (\mathbb{Z} - \{$ n points $\})^l \times \mathbb{R}^{lm}$ are precisely in one to one correspondence with the integer points in $\mathbb{Z}^l \times (\mathbb{Z} - \{$ n points $\})^l$. Therefore, we can prove the theorem if we can show that $B_R(0) \cap \left( \mathbb{Z}^l \times (\mathbb{Z} - \{$ n points $\})^l \right)$ grows exponentially as $l$ gets bigger and bigger. This follows from Lem. A.3. $\qquad\square$

We now move on to give the proof of Thm. 5.3. This will be done by using some standard lemmas from Fourier analysis.

**Lemma A.4.** *Let $f \in L^2(\mathbb{R})$ be a band-limited signal with maximum frequency $\omega_{\max}$. Suppose we sample $f$ at the points $f(nT)$ where $\frac{1}{T} \geq 2\omega_{\max}$. Let*

$$F_N(x) = \sum_{i=-N}^{N} f(nT) \operatorname{sinc}\left(\frac{1}{T}(x - nT)\right). \tag{44}$$

*Then we have the bound*

$$|f - F_N| \leq \mathcal{O}\left(\frac{1}{\sqrt{N}}\right) \tag{45}$$

The proof of this lemma can be found in Olson (2017).

**Lemma A.5.** *The set of band-limited functions denoted $\mathcal{B}$ is a dense set of $L^2(\mathbb{R})$. This means given any signal $f \in L^2(\mathbb{R})$ and any threshold $\epsilon > 0$ we can find a band-limited function $g \in \mathcal{B} \subseteq L^2(\mathbb{R})$ such that*

$$||f - g||_{L^2(\mathbb{R})} < \epsilon. \tag{46}$$

**Lemma A.6.** *A band-limited function $f \in \mathcal{B}$ is necessarily analytic on the whole real line and thus continuous on the whole real line.*

*Proof of Thm. 5.3.* The proof of Thm. 5.3 proceeds as follows. We first use Lem. A.5 to find a $g \in \mathcal{B}$ such that $||f - g||_{L^2(\mathbb{R})} < \frac{\epsilon}{4}$. By Lem. A.6 we have that $g$ is necessarily continuous and by assumption we have that $f$ is continuous. Therefore, we can choose $g$ so that

$$g(x_i) = f(x_i) \text{ for } 1 \leq i \leq n. \tag{47}$$

Then using the fact that $L^2$-convergence implies pointwise convergence (Stein & Shakarchi, 2009), we have that for any $x \in [0, 1]$ it holds $|f - g| < \frac{\epsilon}{4}$.

The next step is to establish the theorem for the bandlimited function $g$. Denote the maximum frequency present in $g$ by $\omega_{max}$. We then choose a collection of points $\{p_i\}_{i \in \mathbb{Z}}$ whose distance between successive elements $|p_{i+1} - p_i| = T$ where $\frac{1}{T} \geq 2\omega_{max}$ and such that $\{x_i\}$ are contained within $\{p_i\}$. Fig. 8 gives a pictorial representation of how the points $\{x_i\}$ will look within $\{p_i\}$.

We then consider the sum

$$\sum_{i=-N}^{N} g(p_i) \operatorname{sinc}\left(\frac{1}{T}(x - p_i)\right) \tag{48}$$

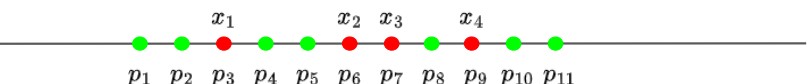

Figure 8: The original data points $\{x_i\}$ are shown in red. The green points are the new equally spaced points $\{p_i\}$ that contain $\{x_i\}$.

and note that this sum is precisely the first $2N$ terms of the Nyquist-Shannon-Whittaker (NSW) series.

Then observe that using the same technique as in the proof of Thm. 5.2, we can represent this sum as a shallow sinc activated neural network $\mathcal{N}$ with $2N$ neurons in the hidden layer. That is, there exists a parameter $\theta$, given by the proof of Thm. 5.2, such that

$$\mathcal{N}(\theta; x) = \sum_{i=-N}^{N} g(p_i) \operatorname{sinc}\left(\frac{1}{T}(x - p_i)\right). \tag{49}$$

and hence

$$\mathcal{N}(\theta; p_i) = g(p_i). \tag{50}$$

If we choose, $N$ large and in particular larger than $n$ we can then show that two properties of $\mathcal{N}$ must hold. First that since $\{x_i\}_{i=1}^{n}$ are contained within $\{p_i\}$ we must have that

$$\mathcal{N}(\theta; x_i) = g(x_i) = f(x_i) \tag{51}$$

as $g = f$ on the data set by construction. Then applying Lem. A.4, we can find an $N >> 1$ very large such that

$$|g(x) - \mathcal{N}(\theta; x)| < \frac{C}{\sqrt{N}} < \frac{\epsilon}{4} \tag{52}$$

for all $x \in [0, 1]$ and for some fixed $C > 0$.

Suppose we now add $q$ neurons to the hidden layer of $\mathcal{N}$. Then $\mathcal{N}$ takes the form

$$\mathcal{N}(\theta; x) = \sum_{i=-N}^{N} g(p_i) \operatorname{sinc}\left(\frac{1}{T}(x - p_i)\right) + \sum_{i=1}^{q} c_i \operatorname{sinc}(a_i x + b_i) \tag{53}$$

where the $a_i$, $b_i$ and $c_i \in \mathbb{R}$ are the extra parameters coming from adding $q$ neurons. Observe that the points $\{p_i\}$ are all equally spaced of distance $T$. Hence we can write

$$x_i = m_i T \tag{54}$$

for some $m_i \in \mathbb{Z}$. Then in order for $\mathcal{N}$ to satisfy the labels $f(x_i)$ on the set $\{x_i\}$ we can choose $a_i \in \frac{1}{T}\mathbb{Z}$ i.e. $a_i = \frac{n_i}{T}$ for any $n_i \in \mathbb{Z}$ and $b_i \in \mathbb{Z} - \{a_i m_1, \ldots, a_i m_n\}$ and $c_i \in \mathbb{R}$. Using the fact that $\operatorname{sinc}(m) = 0$ for $m \in \mathbb{Z}\backslash\{0\}$ and letting $\theta^*$ denote the parameters with $a_i$, $b_i$ and $c_i$ satisfying the above mentioned constraints that

$$\mathcal{N}(\theta^*; x_i) = f(x_i) \tag{55}$$

implying that all these new parameters $\theta^*$ are a global minimum for the loss function

$$\mathcal{L}_2(\theta) = \frac{1}{2n} \sum_{i=1}^{n} \left(\mathcal{N}(\theta; x_i) - f(x_i)\right)^2. \tag{56}$$

We then find that with these new parameters $\theta^*$ that

$$|f(x) - N(\theta^*; x)| \le |f(x) - g(x)| + |g(x) - N(\theta^*; x)| \tag{57}$$

for any $x \in [0, 1]$. We already know that $|f(x) - g(x)| < \frac{\epsilon}{4}$ so in order to prove the theorem it suffices to prove that

$$|g(x) - N(\theta^*; x)| < \frac{\epsilon}{2} \tag{58}$$

for any $x \in [0, 1]$.

In order to do this we observe that we can write

$$|g(x) - N(\theta^*; x)| \leq \left| g(x) - \sum_{i=-N}^{N} g(p_i) \operatorname{sinc}\left(\frac{1}{T}(x - p_i)\right) \right| + \left| \sum_{i=1}^{q} c_i \operatorname{sinc}(a_i x + b_i) \right| \quad (59)$$

where we remind that reader that $a_i \in \frac{1}{T}\mathbb{Z}$, $b_i \in \mathbb{Z}\backslash\{a_i m_1, \ldots, a_i m_n\}$ and $c_i \in \mathbb{R}$. We already chose $N$ large so that

$$\left| g(x) - \sum_{i=-N}^{N} g(p_i) \operatorname{sinc}\left(\frac{1}{T}(x - p_i)\right) \right| + \left| < \frac{\epsilon}{4} \right. \quad (60)$$

see equation 52. Therefore, we just need to bound the term

$$\left| \sum_{i=1}^{q} c_i \operatorname{sinc}(a_i x + b_i) \right|. \quad (61)$$

If we can show that this is less than $\frac{\epsilon}{4}$ we are done. To do this we observe that for any $x \in \mathbb{R}$ $|\operatorname{sinc}(x)| \leq 1$. Therefore

$$\left| \sum_{i=1}^{q} c_i \operatorname{sinc}(a_i x + b_i) \right| \leq \sum_{i=1}^{q} |c_i|. \quad (62)$$

Write $c_i = \lambda$ for some $\lambda \in \mathbb{R}$ so that the sum becomes

$$\sum_{i=1}^{q} |c_i| = q\lambda. \quad (63)$$

We then have that $\lambda$ must satisfy the constraint

$$\lambda \in \left( -\frac{\epsilon}{4q}, \frac{\epsilon}{4q} \right). \quad (64)$$

With $\lambda$ satisfying this constraint we find that the parameters $\theta^*$ such that $a_i \in \frac{1}{T}\mathbb{Z}$ and $b_i \in \mathbb{Z}\backslash\{a_i m_1, \ldots, a_i m_n\}$ and $c_i = \lambda \in \left( -\frac{\epsilon}{4q}, \frac{\epsilon}{4q} \right)$ must satisfy the bound

$$|f(x) - \mathcal{N}(\theta^*; x)| < \epsilon. \quad (65)$$

Furthermore, all these parameters $\theta^*$ are global minima of the loss function

$$\mathcal{L}_2(\theta) = \frac{1}{2n} \sum_{i=1}^{n} \left( \mathcal{N}(\theta; x_i) - f(x_i) \right)^2. \quad (66)$$

We thus see that these generalizable global minima are parameterized by the set $\frac{1}{T}\mathbb{Z} \times \left( -\frac{\epsilon}{4q}, \frac{\epsilon}{4q} \right)$ and correspond to distinct global minimum loss valleys with the different loss valleys parameterized by $\frac{1}{T}\mathbb{Z}$.

We therefore see that we have added a total of $N + q - n$ new neurons to the original $\mathcal{N}$ that had $n$ neurons in the hidden layer. This shows that as long as we take $l \geq N - n$ and then adding $l$ neurons to $\mathcal{N}$ gives the result of the theorem. This provides a quantitative bound on how large $l$ needs to be in order to get the result of the theorem. $\square$

### A.1.2 RESULTS FOR DEEP SINC NETWORKS

In Sec. 5.1 we stated our main Thm. 5.2 which deals with how the global minima in the loss landscape of the $\ell^2$ loss $\mathcal{L}_2$, see equation 5, changes as we add more neurons to the hidden layer of our network. Another way of adding more parameters to a network is to add another hidden layer i.e. add more depth. In this section we show that our Thm. 5.2 has a generalization to the case of deep sinc activated networks.

We start with some propositions.

**Proposition A.7.** *Let $X = \{(x_i, y_i)\}_{i=1}^n$ be a labelled data set. Let $\mathcal{N}(x; \theta)$ be a* sinc *activated shallow neural network with $n$ neurons. Let*

$$\mathcal{L}_2(\theta) := \frac{1}{2n} \sum_{i=1}^n (\mathcal{N}(x_i; \theta) - y_i)^2 \tag{67}$$

*denote the $\ell^2$ loss objective function.*

*Then adding $1$ extra hidden layer of $n$ neurons to $\mathcal{N}(x; \theta)$ results in an increase in global minimum valleys of $\mathcal{L}_2$ parameterized by the set $\mathbb{Z}^{n(n-1)} \times \mathbb{R}^n$. Furthermore, we can write down explicit expressions for each of these new global minima.*

*Proof.* To begin with we assume the data is one dimensional i.e. $x_i \in \mathbb{R}$ and $y_i \in \mathbb{R}$. Furthermore, we assume that our data points are normalized so that $x_i = \frac{p_i}{q}$ for $p_i, q \in \mathbb{Z}$.

When we add an extra hidden layer with $n$ neurons to the network $\mathcal{N}$, we obtain a deep network with 2 hidden layers. We will denote the weights and biases of each layer of this new network as follows. The first hidden layer will have weights and biases denoted by $(W_1, b_1)$, the second hidden layer by $(W_2, b_2)$ and the output layer by $(W_3, b_3)$. The dimensions of these weights and biases will be $W_1$ and $b_1$ will be $n \times 1$, $W_2$ will be an $n \times n$ matrix and $b_2$ will be $n \times 1$. Finally, $W_3$ will be $1 \times n$ and $b_3$ will be $1 \times 1$.

The extra global minima that arise from adding one extra hidden layer will arise from the global minimum found in Prop. A.1. Thus the weight $W_1$ and bias $b_1$ will be given by

$$W_1 = [q, \ldots, q]^T \text{ and } b_1 = [-p_1, \ldots, -p_n]^T. \tag{68}$$

For now we will write the weight $W_2$ and bias $b_2$ as

$$W_2 = \begin{bmatrix} w_{11}^2 & \cdots & w_{1n}^2 \\ \vdots & \vdots & \vdots \\ w_{n1}^2 & \cdots & w_{nn}^2 \end{bmatrix} \text{ and } b_2 = [b_1^2, \ldots, b_n^2]^T \tag{69}$$

and the weight $W_3$ and bias $b_3$ as

$$W_3 = [w_1^3, \ldots, w_n^3] \text{ and } b_3 = b. \tag{70}$$

With the weights and biases defined above denoted by $\theta$ the structure of the network takes the following form

$$\mathcal{N}(\theta, x) = w_1^3 \text{sinc}(w_{11}^2 \text{sinc}(qx - p_1) + \cdots + w_{1n}^2 \text{sinc}(qx - p_n) + b_1^2) \tag{71}$$
$$+$$
$$\vdots$$
$$+$$
$$w_n^3 \text{sinc}(w_{n1}^2 \text{sinc}(qx - p_1) + \cdots + w_{nn}^2 \text{sinc}(qx - p_n) + b_n^2) + b.$$

If we let $b = 0$, $w_i^3 = y_i$ for $1 \leq i \leq n$, $b_i^2 = -w_{ii}^2$, and then allow $w_{ii}^2 \in \mathbb{R}$ and $w_{ij}^2 = n_{ij} - b_j^2$ for $n_{ij} \in \mathbb{Z}$ and for $i \neq j$. We see that any parameter $\theta^*$ that satisfies these constraints satisfies the following

$$\mathcal{N}(\theta^*; x_i) = y_i \text{ for } 1 \leq i \leq n \tag{72}$$

which follows from that fact that $\text{sinc}(m) = 0$ for any $m \in \mathbb{Z} - \{0\}$ and $\text{sinc}(0) = 1$. It thus follows that such parameters are global minima of the loss function $\mathcal{L}_2$.

We therefore see that the extra global minima that arise from adding one hidden layer with $n$ neurons can be parameterized by $\mathbb{R}^n \times \mathbb{Z}^{n(n-1)}$ and hence are global minimum valleys.

The next step is to consider the case that the data $x_i \in \mathbb{R}^k$ and $y_i \in \mathbb{R}$. We write the data as follows:

$$x_1 = [x_{11}, \ldots, x_{k1}]^T$$
$$\vdots$$
$$x_n = [x_{1n}, \ldots, x_{kn}]^T.$$

As in the one dimensional case, we write each data coordinate over a common denominator so that $x_{ij} = \frac{p_{ij}}{q}$.

Then we define a parameter $\theta^*$ for a shallow sinc activated neural network $\mathcal{N}$ as follows: The weight $W_1$ and bias $b_1$ of the hidden layer will be:

$$W_1 = \begin{bmatrix} q & \cdots & q \\ \vdots & \vdots & \vdots \\ q & \cdots & q \end{bmatrix} \text{ and } b_1 = \begin{bmatrix} -(p_{11}+ & \cdots & +p_{k1}) \\ \vdots & \vdots & \vdots \\ -(p_{1n}+ & \cdots & +p_{kn}) \end{bmatrix} \tag{73}$$

Note that in this case $W_1$ has dimensions $n \times k$ and $b_1$ $n \times 1$. The weight $W_2$ and bias $b_2$ of the second hidden layer will be defined just in the same way as above in the case that the data was assumed one dimensional and similarly for the weight $W_3$ and bias $b_3$ of the output layer. We therefore, see that once again with such weights and biases $\mathcal{N}(\theta^*; x_i) = y_i$, which means such parameters are global minima of the loss function $\mathcal{L}_2$ and these extra global minima are parameterized by $\mathbb{Z}^{n(n-1)} \times \mathbb{R}^n$ and hence are global minimum valleys.

Finally, for the case that $x_i \in \mathbb{R}^k$ and $y_i \in \mathbb{R}^m$ the proof follows the strategy of Prop. A.1.

In this case we write $y_i = (y_{1i}, \ldots, y_{mi})$, for $1 \le i \le n$. We will also use the same convention we used for the data points $x_i$ above.

In this case the weights and bias of the first hidden layer will be the same as in equation 73. The weights $W_2$ and bias $b_2$ for the second hidden layer will be exactly the same as those found for the case the data was assumed one dimensional. The weight $W_3$ and bias $b_3$ for the output layer will be defined by

$$W_2 = \begin{bmatrix} y_{11} & \cdots & y_{1n} \\ \vdots & \vdots & \vdots \\ y_{m1} & \cdots & y_{mn} \end{bmatrix} \text{ and } b_2 = [0, \cdots, 0]^T. \tag{74}$$

For an arbitrary input $z = [z_1, \ldots, z_k]^T$ we have

$$\mathcal{N}(z; \theta^*) = [\mathcal{N}_1(z; \theta^*), \ldots, \mathcal{N}_m(z; \theta^*)]^T \tag{75}$$

where

$$\mathcal{N}_j(z; \theta^*) = y_{j1}\text{sinc}\bigg( qz_1 - p_{11} + \cdots + qz_k - p_{k1} \bigg) + \cdots$$

$$+ y_{jn}\text{sinc}\bigg( qz_1 - p_{1n} + \cdots + qz_k - p_{kn} \bigg)$$

for $1 \le j \le m$.

It is clear from this that

$$\mathcal{N}(x_i; \theta^*) = y_i \text{ for all } 1 \le i \le n. \tag{76}$$

This gives an explicit representation of $\theta^*$ in this setting and shows that it is a global minimum for the loss function $\mathcal{L}_2$. Furthermore, once again we have that the extra global minima are parameterized by $\mathbb{R}^n \times \mathbb{Z}^{n(n-1)}$ and are valleys. $\qquad \square$

**Proposition A.8.** *Let $X = \{(x_i, y_i)\}_{i=1}^n$ be a data set with $x_i \in \mathbb{R}^k$ and $y_i \in \mathbb{R}^m$. Let $\mathcal{N}(x; \theta)$ be a sinc activated shallow neural network with $n$ neurons given by Prop. A.7. Let*

$$\mathcal{L}_2(\theta) := \frac{1}{2n} \sum_{i=1}^n (\mathcal{N}(x_i; \theta) - y_i)^2 \tag{77}$$

*denote the $\ell^2$ loss objective function.*

*Then adding $l$ extra hidden layers with $n$ neurons each to $\mathcal{N}(x; \theta)$ results in an increase in global minima of $\mathcal{L}_2$ parameterized by the set $\mathbb{Z}^{ln(n-1)} \times \mathbb{R}^{ln}$. Furthermore, we can write down explicit expressions for each of these new global minima.*

*Proof.* The proof of this proposition follows exactly the same approach of Prop. A.7. One simply uses induction on the number of hidden layers with the base case being Prop. A.7. $\qquad \square$

The following theorem is the analogue of Thm. 5.2 for the case of adding parameters by adding depth.

**Theorem A.9.** *Let $X = \{(x_i, y_i)\}_{i=1}^n$ be a data set with $x_i \in \mathbb{R}^k$ and $y_i \in \mathbb{R}^m$. Let $\mathcal{N}(x; \theta)$ be a shallow neural network with $n$ neurons. Let*

$$\mathcal{L}_2(\theta) := \frac{1}{2n} \sum_{i=1}^n (\mathcal{N}(x_i; \theta) - y_i)^2 \tag{78}$$

*denote the $\ell^2$ loss function. Let $\mathcal{G}_R(n, 1)$ denote the number of distinct global minimum valleys of $\mathcal{L}_2$ in the ball $B_R(0)$ of radius $R \geq 1$ around the origin $0$ where the dependence of $n$ comes from the $n$ neurons of $\mathcal{N}$ and the $1$ denotes that $\mathcal{N}$ has $1$ hidden layer. Then if we add $l$ hidden layers, each with $n$ neurons, to $\mathcal{N}$ we have that $\mathcal{G}_R(n, l)$ grows at least exponentially in $l$.*

*Proof.* From Prop. A.8 we see that when we add $l$ hidden layers, each with $n$ neurons, there are extra global minima for the objective function $\mathcal{L}_2$ that are parameterized by $\mathbb{R}^{nl} \times \mathbb{Z}^{ln(n-1)}$. Each of these global minimum valleys are parameterized by $\mathbb{Z}^{ln(n-1)}$. Thus we see that the number of such valleys grows like the number of integer points in $B_R(0) \cap \mathbb{Z}^{ln(n-1)}$, which has exponential growth in $l$ by Lem. A.3. $\qquad\square$

### A.1.3 Results for shallow ReLU networks

In this section we want to give the proof of Thm. 5.5. In order to do so we will need to establish a correspondance between $ReLU$ shallow networks and the Triangle function T.

The starting point is Lem. 5.4 whose proof we now give.

*Proof of Lem. 5.4.* The proof of equation 17 in Lem. 5.4 follows immediately from the definition of the ReLU function.

The parameter $\theta^*$ is defined as follows. The weight $W_1$ and bias $b_1$ of the hidden layer are defined by

$$W_1 = [1, 1, 1]^T \text{ and } [1, -1, 0]^T. \tag{79}$$

The weight $W_2$ and bias $b_2$ of the output layer are defined by

$$W_2 = [1, 1, -2] \text{ and } b_2 = 0. \tag{80}$$

Using these parameters we see that

$$\mathcal{N}(x; \theta^*) = \text{ReLU}(x + 1) + \text{ReLU}(x - 1) - 2\text{ReLU}(x) \tag{81}$$
$$= T(x) \text{ by } equation\ 17. \tag{82}$$

For the case of $T(\omega(x - a))$ we have

$$W_1 = [\omega, \omega, \omega]^T \text{ and } b_1 = [1 - \omega a, -1 - \omega a, -\omega a] \tag{83}$$

and $W_2$ and $b_2$ the same as above. $\qquad\square$

Sampling with the triangle function leads to a piecewise linear interpolant as shown in the following lemma.

**Lemma A.10.** *Let $f \in L^2(\mathbb{R})$ and let $T : \mathbb{R} \to \mathbb{R}$ denote the triangular function defined by $T(x) = \max\{1 - |x|, 0\}$. Suppose we sample the signal $f$ at the integer points, $f(n)$ for $n \in \mathbb{Z}$. Then the series*

$$s(x) = \sum_{n=-\infty}^{\infty} f(n) T(x - n) \tag{84}$$

*is a piecewise linear interpolation of the signal $f$.*

*Proof.* The starting point is to observe that the triangular function is linear on the regions $[-1, 0]$ and $[0, 1]$ and completely zero outside these regions. This means that in the summation equation 84

the only non-zero term will come from the nearest sampled points $n_1 \leq x \leq n_2$ where $n_1 = \lfloor x \rfloor$ and $n_2 = \lceil x \rceil$. Thus for $n_1 \leq x \leq n_2$ we see that the summation breaks down to

$$s(x) = f(n_1)T(x - n_1) + f(n_2)T(x - n_2). \tag{85}$$

Applying the definition $T(x) = \max\{1 - |x|, 0\}$ we find

$$T(x - n_1) = 1 - (x - n_1) = n_2 - x \tag{86}$$
$$T(x - n_2) = 1 - (n_2 - x) = x - n_1. \tag{87}$$

This then shows that for $n_1 \leq x \leq n_2$

$$s(x) = f(n_1)(n_2 - x) + f(n_2)(x - n_1) \tag{88}$$

which is precisely the formula for the linear interpolation between $f(n_1)$ and $f(n_2)$. Hence we see that equation 84 is a piecewise linear approximation to the signal $f$ as required. $\square$

In the above Lem. A.10 we assumed the signal $f$ was sampled on the integers. In general, in the case that the signal is sampled on a discrete set of equally spaced points $\{x_i\}$ such that $|x_i - x_j| = d$ we get an equivalent lemma by using the scaled triangular function given by $T_d(x) = T(\frac{1}{d}x)$. By taking the sampling points $\{x_i\}$ closer together one obtains a better linear approximation of $f$.

Lem. 5.4 and A.10 give another way to see that ReLU networks perform piecewise linear interpolation.

Before we can prove the main Thm. 5.5 from Sec. 5.2 we will state and prove some propositions.

**Proposition A.11.** *Let $X = \{(x_i, y_i)\}_{i=1}^n$ be a data set with $x_i \in \mathbb{R}^k$ and $y_i \in \mathbb{R}^m$. Let $\mathcal{N}(x; \theta)$ be a* ReLU *activated shallow neural network with $3n$ neurons. Then there exists a parameter $\theta^*$ such that*

$$\mathcal{N}(x_i; \theta^*) = y_i \text{ for all } 1 \leq i \leq n. \tag{89}$$

*In particular, $\theta^*$ is a global minimum for the $\ell^2$ loss objective*

$$\mathcal{L}_2(\theta) := \frac{1}{6n} \sum_{i=1}^{3n} (\mathcal{N}(x_i; \theta) - y_i)^2. \tag{90}$$

*Furthermore, we can write down an explicit expression for the parameter $\theta^*$.*

*Proof.* The proof of this theorem follows the sampling strategy undertaken in the proof of Prop. A.1 with the use of Lem. A.10.

To begin with we assume $x_i \in \mathbb{R}$ and $y_i \in \mathbb{R}$. Choose $\epsilon_1, \ldots, \epsilon_n > 0$ so that

$$T(\frac{1}{\epsilon_i}(x_j - x_i) = 0 \text{ for all } i \neq j. \tag{91}$$

We then define the parameter $\theta^*$ as follows. The weight $W_1$ and bias $b_1$ of the hidden layer will be

$$W_1 = \left[ \frac{1}{\epsilon_1}, \frac{1}{\epsilon_1}, \frac{1}{\epsilon_1}, \frac{1}{\epsilon_2}, \frac{1}{\epsilon_2}, \frac{1}{\epsilon_2}, \ldots, \frac{1}{\epsilon_n}, \frac{1}{\epsilon_n}, \frac{1}{\epsilon_n} \right]^T \tag{92}$$

$$b_1 = \left[ \frac{-x_1}{\epsilon_1} + 1, \frac{-x_1}{\epsilon_1} - 1, \frac{-x_1}{\epsilon_1}, \ldots, \frac{-x_n}{\epsilon_n} + 1, \frac{-x_n}{\epsilon_n} - 1, \frac{-x_n}{\epsilon_n} \right]^T. \tag{93}$$

The output layer will have weight $W_2$ and bias $b_2$ given by

$$W_2 = \left[ y_1, y_1, -2y_1, y_2, y_2, -2y_2, \ldots, y_n, y_n, -2y_n \right] \tag{94}$$

$$b_2 = 0. \tag{95}$$

We then find that

$$\mathcal{N}(x; \theta^*) = \sum_{i=1}^n y_i T\left( \frac{1}{\epsilon_i}(x - x_i) \right) \tag{96}$$

which implies that

$$\mathcal{N}(x_i; \theta^*) = y_i \tag{97}$$

by the choice of numbers $\epsilon_1, \ldots, \epsilon_n$ and the definition of the triangular function $T$.

The general case of data $\{(x_i, y_i)\}_{i=1}^n$ with $x_i \in \mathbb{R}^k$ and $y_i \in \mathbb{R}^m$ follows the exact same strategy as we did in the proof for Prop. A.1. $\qquad\square$

**Proposition A.12.** *Let $X = \{(x_i, f(x_i))\}_{i=1}^n$ be a data set with $x_i \in \mathbb{R}^k$ and $y_i \in \mathbb{R}^m$. Let $\mathcal{N}(x; \theta)$ be a* ReLU *activated shallow neural network with $3n$ neurons given by the above theorem. Let*

$$\mathcal{L}_2(\theta) := \frac{1}{6n} \sum_{i=1}^{3n} (\mathcal{N}(x_i; \theta) - y_i)^2 \tag{98}$$

*denote the $\ell^2$ loss objective function.*

*Then adding 3 extra neurons to the hidden layer of $\mathcal{N}(x; \theta)$ results in an increase in global minima valleys of $\mathcal{L}_2$ parameterized by the set*

$$\left( \mathbb{R} - \{ n \text{ closed intervals} \} \right) \times \left( \mathbb{R} - \{ n \text{ closed intervals} \} \right) \times \mathbb{R}^m. \tag{99}$$

*Proof.* The poof of this follows the approach taken in Prop. A.11. In Prop. A.11, we found a parameter $\theta^*$ that was a global minimum when our network had $3n$ neurons. We will build the new global minima from the representation of $\theta^*$ found in Prop. A.11. We will start with the simpler case of one dimensional data. So assume that $x_i \in \mathbb{R}$ and the labels $y_i \in \mathbb{R}$.

Let us represent the weight $W_1$ and bias $b_1$ of the hidden layer by

$$W_1 = \left[ \frac{1}{\epsilon_1}, \frac{1}{\epsilon_1}, \frac{1}{\epsilon_1}, \ldots, \frac{1}{\epsilon_n}, \frac{1}{\epsilon_n}, \frac{1}{\epsilon_n}, a, a, a \right]^T \tag{100}$$

$$b_1 = \left[ \frac{-x_1}{\epsilon_1} + 1, \frac{-x_1}{\epsilon_1} - 1, \frac{-x_1}{\epsilon_1}, \ldots, \frac{-x_n}{\epsilon_n} + 1, \frac{-x_n}{\epsilon_n} - 1, \frac{-x_n}{\epsilon_n}, -a\lambda + 1, a\lambda - 1, a\lambda \right]^T \tag{101}$$

where $a, \lambda \in \mathbb{R}$. We represent the weight $W_2$ and bias $b_2$ of the output layer by

$$W_2 = \left[ y_1, y_1, -2y_1, \ldots, y_n, y_n, -2y_n, c, c, -2c \right] \tag{102}$$

$$b_2 = 0 \tag{103}$$

for $c \in \mathbb{R}$. We then see that for such parameters we have that

$$\mathcal{N}(x; \theta) = \sum_{i=1}^n y_i T\left( \frac{1}{\epsilon_i}(x - x_i) \right) + cT(a(x - \lambda)). \tag{104}$$

We then observe that $\mathcal{N}(x; \theta)$ will fit the training data provided the term

$$cT(a(x_i - \lambda)) = 0 \text{ for } 1 \leq i \leq n. \tag{105}$$

Mathematically this will happen provided $\lambda$ lies outside the closed intervals $[x_i - \epsilon_i, x_i + \epsilon_i]$ for all $1 \leq i \leq n$ and

$$|a| > \max\{|\lambda - (x_i + \epsilon)|, |\lambda - (x_i - \epsilon)|\} \text{ for all } 1 \leq i \leq n. \tag{106}$$

This constraint can be encoded by allowing

$$\lambda \in \mathbb{R} - \bigcup_{i=1}^n [x_i - \epsilon_i, x_i + \epsilon_i] \tag{107}$$

$$a \in \mathbb{R} - \bigcup_{i=1}^n \left[ -(\lambda - (x_1 - \epsilon_i)), \lambda - (x_1 - \epsilon_i) \right] \tag{108}$$

$$c \in \mathbb{R}. \tag{109}$$

Letting $\theta^*$ be determined by weights and biases satisfying the above constraint, we see that

$$\mathcal{N}(\theta^*; x_i) = y_i \text{ for } 1 \leq i \leq n \tag{110}$$

hence with such parameters we see that

$$\mathcal{L}_2(\theta^*) = 0 \tag{111}$$

showing that the extra parameters $\theta^*$ are all global minima.

From what we showed above we can see these extra global minima are parameterized by the set

$$\left( \mathbb{R} - \bigcup_{i=1}^{n} [x_i - \epsilon_i, x_i + \epsilon_i] \right) \times \left( \mathbb{R} - \bigcup_{i=1}^{n} \left[ -(\lambda - (x_1 - \epsilon_i)), \lambda - (x_1 - \epsilon_i) \right] \right) \times \mathbb{R} \tag{112}$$

The general case of data $x_i \in \mathbb{R}^k$ with labels $y_i \in \mathbb{R}^m$ uses the above together with the exact same approach we took for Prop. A.2. $\qquad\square$

We are now in a position to prove Thm. 5.5.

*Proof of Thm. 5.5.* The proof of this theorem proceeds by inducting over $l$. The base case is given by Prop. A.12. Assuming the statement is true for $l - 1$ for $l > 0$ we can run through the proof of Prop. A.12 and see that by adding another 3 neurons we get extra global minima parameterized by the set

$$\left( \mathbb{R} - \bigcup_{i=1}^{n} [x_i - \epsilon_i, x_i + \epsilon_i] \right) \times \left( \mathbb{R} - \bigcup_{i=1}^{n} \left[ -(\lambda - (x_1 - \epsilon_i)), \lambda - (x_1 - \epsilon_i) \right] \right) \times \mathbb{R}^m. \tag{113}$$

Combined with the induction step this yields the statement of the theorem.

Observe that as we add $3l$ neurons the dimension of the global minimum valley given by the above scales in dimension by $l$ showing that overparameterization leads to higher dimensional global minimum loss valleys that scale at least linearly in $l$. $\qquad\square$

In the case of shallow ReLU networks we have an analogue of Thm. 5.3. In order to derive such a theorem we start with the following simple lemma from linear interpolation Stein & Shakarchi (2009).

**Lemma A.13.** *Let $f \in L^2(\mathbb{R})$ then for any given $\epsilon > 0$, there exists a function $g \in L^($\mathbb{R})$ such that $g$ is a continuous piecewise linear function and such that $||f - g||_{L^2(\mathbb{R})} < \epsilon$.*

**Theorem A.14.** *Let $f \in L^2(\mathbb{R})$ be a continuous signal, and let $\epsilon > 0$ be a fixed threshold. Consider a dataset $(x_i, f(x_i))_{i=1}^{n}$ obtained by sampling $f$. Let $\mathcal{N}(\theta; x)$ be a shallow feedforward network with ReLU activation and $3n$ neurons in its hidden layer. Define the $\ell^2$ loss function based on the parameters $\theta$ of $\mathcal{N}$ as follows:*

$$\mathcal{L}_2(\theta) := \frac{1}{6n} \sum_{i=1}^{3n} \left( \mathcal{N}(\theta; x_i) - f(x_i) \right)^2. \tag{114}$$

*If we add $l > 0$ neurons to the hidden layer of $\mathcal{N}$, for sufficiently large $l$, there are an infinite number of parameters $\theta$ lying in a global minimum valley that satisfy the following bound:*

$$|f(x) - \mathcal{N}(\theta^*; x)| < \epsilon \tag{115}$$

*for any $x \in [0, 1] \setminus \{x_i\}_{i=1}^{n}$.*

*Proof.* The proof of Thm. A.14 proceeds in a similar way to the proof of Thm. 5.3 with one key difference. We will use the triangle function as the sampling kernel. We first use Lem. A.13 to find a $g \in L^2(\mathbb{R})$ such that $||f - g||_{L^2(\mathbb{R})} < \frac{\epsilon}{4}$ and such that $g$ is a coninutous piecewise linear curve. Therefore, we can choose $g$ so that

$$g(x_i) = f(x_i) \text{ for } 1 \leq i \leq n. \tag{116}$$

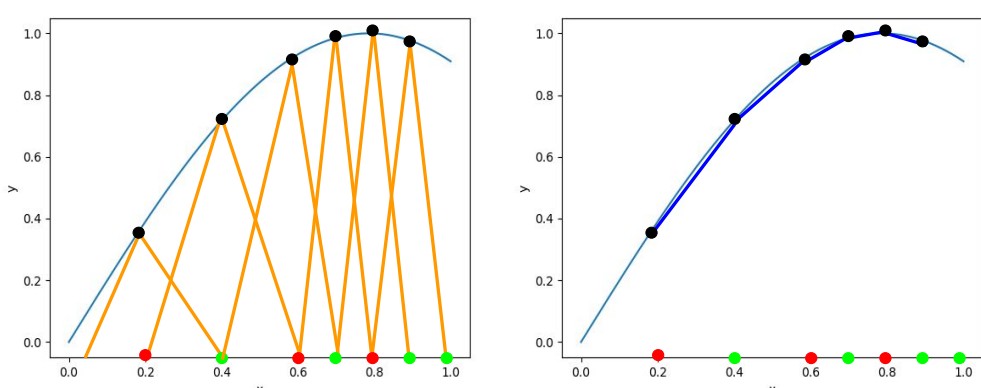

Figure 9: Left: The original function in blue is sampled at the black points with shifted triangle functions at the centre of each sampled point. Right: The reconstruction on taking the sum of the shifted triangular functions produces a continuous piecwise linear approximation to the original function.

Then using the fact that $L^2$-convergence implies pointwise convergence (Stein & Shakarchi, 2009), we have that for any $x \in [0, 1]$ it holds $|f - g| < \frac{\epsilon}{4}$.

The next step is to establish the theorem for the function $g$. Denote the maximum frequency present in $g$ by $\omega_{max}$. We then choose a collection of points $\{p_i\}_{i \in \mathbb{Z}}$ whose distance between successive elements $|p_{i+1} - p_i| = T$ where $\frac{1}{T} \geq 2\omega_{\max}$ and such that $\{x_i\}$ are contained within $\{p_i\}$. Fig. 8 gives a pictorial representation of how the points $\{x_i\}$ will look within $\{p_i\}$.

The next step is to use Lem. A.10 which implies that sampling with the triangle function T is the same as performing a piecewise linear interpolation that is continuous if the signal being interpolated is continuous. This means, we can find a collection of points $\{p_i\}_{i \in \mathbb{Z}}$ that contains $\{x_i\}$ and some $\epsilon_i > 0$ and $\lambda_i$ and write

$$g(x) = \sum_{i=-\infty}^{\infty} \lambda_i \, \mathrm{T}(\frac{1}{\epsilon_i}(x - p_i)). \tag{117}$$

Fig. 9 gives a schematic viewpoint of how $g$ is constructed from shifted triangle functions.

In particular if we just look at the points $\{p_i\}$ that lie in a small neighbourhood of $[0, 1]$, we have that there is an $N > 0$ such that

$$|g(x) - \sum_{i=-N}^{N} \lambda_i \, \mathrm{T}(\frac{1}{\epsilon_i}(x - p_i))| < \frac{\epsilon}{4} \tag{118}$$

for any $x \in [0, 1]$ and since $g(x_i) = f(x_i)$ we have that the associated $\lambda_i$ will be $f(x_i)$. Furthermore we have that the sum

$$\sum_{i=-N}^{N} \lambda_i \, \mathrm{T}(\frac{1}{\epsilon_i}(x - p_i)) \tag{119}$$

can be represented as a shallow ReLU network $\mathcal{N}(\theta; x)$ with $6N$ neurons in its hidden layer. This follows by using Lem. 5.4. In particular, this implies that

$$\mathcal{N}(\theta; x_i) = f(x_i) \tag{120}$$

so that $\theta$ defines a global minimum of the loss function $\mathcal{L}_2$.

Suppose we now add $3q > 0$ neurons to the hidden layer of $\mathcal{N}$. Then we can write $\mathcal{N}$ with the new parameters from adding these neurons as

$$\sum_{i=-N}^{N} \lambda_i \, \mathrm{T}(\frac{1}{\epsilon_i}(x - p_i)) + \sum_{i=1}^{q} a_i \, \mathrm{T}(\frac{1}{b_i}(x - c_i)). \tag{121}$$

The key observation to make now is that as long as $c_i$ is outside the closed interval $[0,1]$ and $b_i$ is such that the triangle centred at $c_i$ with sides determined by $\frac{1}{b_i}$ and height $a_i$ namely $a_i \, \mathrm{T}(b_i(x-c_i))$ is zero for any point $x \in [0,1]$. We then see that any new parameters arising from adding $q$ neurons to the hidden layer and satisfying these constraints will still satisfy

$$\mathcal{N}(\theta^*; x_i) = f(x_i) \text{ for } 1 \le i \le n \tag{122}$$

and thus the parameters $\theta^*$ are global minima for the loss $\mathcal{L}_2$.

We then see that

$$|f(x) - \mathcal{N}(\theta^*; x)| \le |f(x) - g(x)| + |g(x) - \mathcal{N}(\theta^*; x)| \le \frac{\epsilon}{4} + |g(x) - \mathcal{N}(\theta^*; x)| \tag{123}$$

for any $x \in [0,1]$. The final step is to estimate the quantity $|g(x) - \mathcal{N}(\theta^*; x)|$. We can rewrite this as follows

$$|g(x) - \mathcal{N}(\theta^*; x)| \le |g(x) - \sum_{i=-N}^{N} \lambda_i \, \mathrm{T}(\frac{1}{\epsilon_i}(x - p_i))| + |\sum_{i=1}^{q} a_i \, \mathrm{T}(\frac{1}{b_i}(x - c_i))|. \tag{124}$$

Observe that we already made it so that

$$|g(x) - \sum_{i=-N}^{N} \lambda_i \, \mathrm{T}(\frac{1}{\epsilon_i}(x - p_i))| < \frac{\epsilon}{4}. \tag{125}$$

So we only need to estimate the absolute value of the sum $\sum_{i=1}^{q} a_i \, \mathrm{T}(\frac{1}{b_i}(x - c_i))$. Since $|\mathrm{T}(\frac{1}{b_i}(x - c_i))| \le 1$ we have that

$$|\sum_{i=1}^{q} a_i \, \mathrm{T}(\frac{1}{b_i}(x - c_i))| \le \sum_{i=1}^{q} |a_i|. \tag{126}$$

If we write each $a_i = \eta$ where $\eta \in (-\frac{\epsilon}{4q}, \frac{\epsilon}{4q})$ then

$$\sum_{i=1}^{q} |a_i| < \frac{\epsilon}{4}. \tag{127}$$

We thus see that we need the new parameters $a_i$, $b_i$ and $c_i$ to satisfy the constraints $a_i \in (-\frac{\epsilon}{4q}, \frac{\epsilon}{4q})$, $b_i \in \mathbb{R} \setminus \{\text{closed interval}\}$ and $c_i \in \mathbb{R} \setminus [0,1]$ and with these constraints the parameters $\theta^*$ form a global minimum valley and satisfy the bound

$$|f(x) - \mathcal{N}(\theta^*; x)| < \epsilon \tag{128}$$

for any $x \in [0,1]$.

We therefore see that we have added a total of $N + 3q - 3n$ new neurons to the original $\mathcal{N}$ that had $3n$ neurons in the hidden layer. This shows that as long as we take $l \ge N - 3n$ then adding $l$ neurons to $\mathcal{N}$ gives the result of the theorem. This provides a quantitative bound on how large $l$ needs to be in order to get the result of the theorem. $\qquad \square$

### A.1.4 Results for deep ReLU networks

Thm. 5.5 applies overparameterization by increasing the width of the hidden layer of a shallow neural network. In this section we state and prove results for the case of increasing the depth by adding more hidden layers.

**Proposition A.15.** *Let $X = \{(x_i, f(x_i))\}_{i=1}^{n}$ be a data set with $x_i \in \mathbb{R}^k$ and $y_i \in \mathbb{R}^m$. Let $\mathcal{N}(x; \theta)$ be a* ReLU *activated shallow neural network with $3n$ neurons given by the above theorem. Let*

$$\mathcal{L}_2(\theta) := \frac{1}{6n} \sum_{i=1}^{3n} (\mathcal{N}(x_i; \theta) - f(x_i))^2 \tag{129}$$

*denote the $\ell^2$ loss objective function.*

*Then adding $1$ extra hidden layer of $3n$ neurons to $\mathcal{N}(x; \theta)$ results in an increase in global minima of $\mathcal{L}_2$ parameterized by the set*

$$\mathbb{R}^n \times \left( \mathbb{R} - (-1, 1) \right)^{n(n-1)} \tag{130}$$

*Furthermore, we can write down explicit expressions for each of these new global minima.*

*Proof.* The proof of this proposition will be structured similar to Prop. A.12 with the main idea being that the extra neurons added through the hidden layer can be chosen in such a way that the network still perfectly fits the training data.

Start by assuming the data is one dimensional so that $x_i \in \mathbb{R}$ and $y_i \in \mathbb{R}$ and choose $\epsilon_1, \ldots, \epsilon_n > 0$ so that

$$T(\frac{1}{\epsilon_i}(x_j - x_i) = 0 \text{ for all } i \neq j. \tag{131}$$

We will then build the weights of the extra hidden layer using the weights and biases found in Prop. A.11. Namely, we let

$$W_1 = \left[\frac{1}{\epsilon_1}, \frac{1}{\epsilon_1}, \frac{1}{\epsilon_1}, \frac{1}{\epsilon_2}, \frac{1}{\epsilon_2}, \frac{1}{\epsilon_2}, \ldots, \frac{1}{\epsilon_n}, \frac{1}{\epsilon_n}, \frac{1}{\epsilon_n}\right]^T \tag{132}$$

$$b_1 = \left[\frac{-x_1}{\epsilon_1} + 1, \frac{-x_1}{\epsilon_1} - 1, \frac{-x_1}{\epsilon_1}, \ldots, \frac{-x_n}{\epsilon_n} + 1, \frac{-x_n}{\epsilon_n} - 1, \frac{-x_n}{\epsilon_n}\right]^T. \tag{133}$$

Note that $W_1$ has shape $3n \times 1$ and $b_1$ has shape $3n \times 1$. Let $W_2$ and $b_2$ be the weights and bias of the second hidden layer respectively, and let us notate them as follows

$$W_2 = \begin{bmatrix} w_{1,1}^2 & \cdots & w_{1,3n}^2 \\ \vdots & \vdots & \vdots \\ w_{3n,1}^2 & \cdots & w_{3n,3n}^2 \end{bmatrix} \text{ and } b_2 = [b_1^2, \ldots, b_{3n}^2]^T \tag{134}$$

and the weight $W_3$ and bias $b_3$ as

$$W_3 = [w_1^3, \ldots, w_{3n}^3] \text{ and } b_3 = b. \tag{135}$$

We can then write the network out as

$$\mathcal{N}(\theta, x) = w_1^3 \text{ReLU}\left(w_{1,1}^2 \text{ReLU}(\frac{1}{\epsilon_1}(x - x_1) + 1) + w_{1,2}^2 \text{ReLU}(\frac{1}{\epsilon_1}(x - x_1) + 1) - 2w_{1,3}^2 \text{ReLU}(\frac{1}{\epsilon_1}(x - x_1))\right.$$

$$+ \cdots + w_{1,3n-2}^2 \text{ReLU}(\frac{1}{\epsilon_1}(x - x_1) + 1) + w_{1,3n-1}^2 \text{ReLU}(\frac{1}{\epsilon_1}(x - x_1) + 1)$$

$$\left. - 2w_{1,3n}^2 \text{ReLU}(\frac{1}{\epsilon_1}(x - x_1))\right)$$

$$+$$

$$\vdots$$

$$+$$

$$w_n^3 \text{ReLU}\left(w_{3n,1}^2 \text{ReLU}(\frac{1}{\epsilon_1}(x - x_1) + 1) + w_{3n,2}^2 \text{ReLU}(\frac{1}{\epsilon_1}(x - x_1) + 1) - 2w_{3n,3}^2 \text{ReLU}(\frac{1}{\epsilon_1}(x - x_1))\right.$$

$$+ \cdots + w_{3n,3n-2}^2 \text{ReLU}(\frac{1}{\epsilon_1}(x - x_1) + 1) + w_{3n,3n-1}^2 \text{ReLU}(\frac{1}{\epsilon_1}(x - x_1) + 1)$$

$$\left. - 2w_{3n,3n}^2 \text{ReLU}(\frac{1}{\epsilon_1}(x - x_1))\right) + b$$

If we let $b = 0$, $w_1^3 = y_1$, $w_2^3 = y_1$, $w_3^3 = -2y_1 \ldots$, $w_{3n-2}^3 = y_n$, $w_{3n-1}^3 = y_n$, $w_{3n}^3 = -2y_n$. Then let $w_{i,i}^2 \in \mathbb{R}$ and impose the constraint $|w_{ij}^2 - w_{ii}^2| \geq 1$ for all $j \neq i$. Then observe that any parameter $\theta^*$ satisfying these weight and bias constraints for each layer forced the neural network to satisfy

$$\mathcal{N}(x_i; \theta^*) = y_i \text{ for } 1 \leq i \leq n. \tag{136}$$

For each $i \neq j$ if we write $w_{ij}^2 = r_{ij} + w_{ii}^2$ for $r_{ij} \in \mathbb{R}$. Then the constraint $|w_{ij}^2 - w_{ii}^2| \geq 1$ can be expressed as $|r_{ij}| \geq 1$. Thus we see that the global minima obtained by adding one extra hidden layer can be parameterized by

$$\mathbb{R}^n \times \left(\mathbb{R} - (-1, 1)\right)^{n(n-1)}. \tag{137}$$

The general case where the data is high dimensional so that $x_i \in \mathbb{R}^k$ and $y_i \in \mathbb{R}^m$ follows the exact same proof strategy from Thm. A.9. $\qquad\square$

We can now state and prove the deep analogue of Thm. 5.5 for ReLU networks.

**Theorem A.16.** *Let $X = \{(x_i, y_i)\}_{i=1}^n$ be a data set with $x_i \in \mathbb{R}^k$ and $y_i \in \mathbb{R}^m$. Let $\mathcal{N}(x; \theta)$ be a* ReLU *activated shallow neural network with $3n$ neurons. Let*

$$\mathcal{L}_2(\theta) := \frac{1}{6n} \sum_{i=1}^{3n} (\mathcal{N}(x_i; \theta) - f(x_i))^2 \tag{138}$$

*denote the $\ell^2$ loss objective function.*

*Then adding $l > 0$ extra hidden layers of $3n$ neurons to $\mathcal{N}(x; \theta)$ results in an increase in global minima of $\mathcal{L}_2$ parameterized by the set*

$$\mathbb{R}^{ln} \times \left(\mathbb{R} - (-1, 1)\right)^{ln(n-1)}. \tag{139}$$

*Thus we see that overparameterizing by adding more hidden layers leads to higher dimensional global minimum loss valleys whose dimension grow at worst as $\Omega(l)$.*

*Proof.* The proof of this theorem uses induction on $l$. The base case of $l = 1$ being given by Prop. A.15. The inductive step is then carried out by assuming the theorem is true for $l - 1$ for $l > 1$ and then proceeding with the exact same lines of proof as in Prop. A.15 to obtain that the extra global minima are parameterized by

$$\mathbb{R}^n \times \left(\mathbb{R} - (-1, 1)\right)^{n(n-1)}. \tag{140}$$

Combining this with the induction step leads to the proof of the theorem. □

## A.2 Experiments

### A.2.1 Experimental setup

In this section we discuss the experimental hyperparameters we used for each of the experiments. For each of the four experiments we used the optimizers SGD, Adam, OnePlusOne and L-BFGS.

**SGD:** For this optimizer we used the standard PyTorch implementation with a learning rate of 1e-3 for all experiments. We found we obtained similar results with learning rates of the form 1e-2 and 1e-4 (and lower) but that 1e-3 was the best and seemed to be the commonly used learning rate for SGD in the literature. We observed that with different learning rates overparameterization still led to better train PNSRs and test PSNRs.

**Adam:** For this optimizer we used the standard PyTorch implementation with a learning rate of 1e-4 for all experiments which is what the literature used. We found we obtained similar results with learning rates of the form 1e-2 and 1e-3 (and lower) but that 1e-4 was the best. We observed that with different learning rates overparameterization still led to better train PNSRs and test PSNRs.

**OnePlusOne:** This optimizer operates through an iterative process involving "parents" and "offspring." The algorithm starts with a single solution, referred to as the parent. At each iteration, a new solution, the offspring, is generated by introducing a random mutation to the parent. This mutation typically follows a Gaussian distribution. The offspring is then evaluated based on the objective function. If the offspring achieves a better result than the parent, it replaces the parent; otherwise, the parent remains unchanged. This process continues until convergence or a stopping criterion is met, gradually improving the solution with each iteration. We implemented this optimizer based on the facebook research code available at https://facebookresearch.github.io/nevergrad/. The hyperparameter that needs to be fixed is the number offspring the optimizer sends out to compare against the parent. We found that anywhere between 10 to 30 offspring did the best and thus fixed 20 offspring as our hyperparameter. For each hyperparameter we noticed the trend that oveparameterization yielded better train and test PSNRs.

**L-BFGS:** For this optimizer we used the standard PyTorch implementation with a learning rate of 1e-3 for all experiments. We found we obtained similar results with learning rates of the form 1e-2 and 1e-4 (and lower) but that 1e-3 was the best. We observed that with different learning rates overparameterization still led to better train PNSRs and test PSNRs. We also noticed that the optimizer struggled for the larger experiments image super resolution and the binary occupancy field. This was also found in the literature in work of Saratchandran et al. (2023).

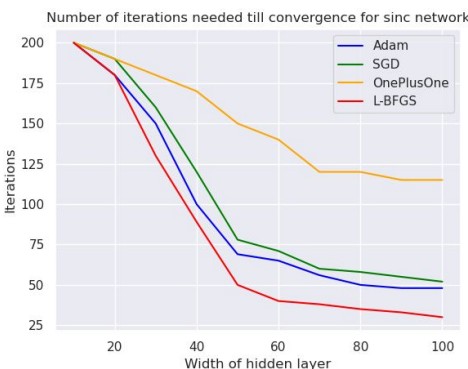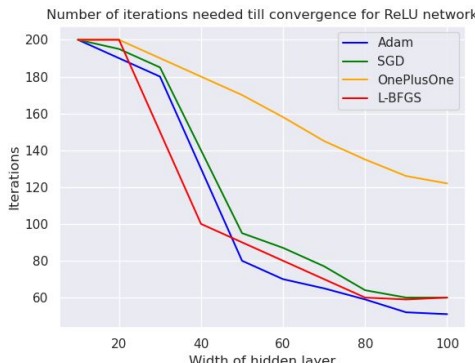

Figure 10: Number of iterations needed to converge for each optimizer as width increases for $\mathrm{sinc}$ (left) and $\mathrm{ReLU}$ networks (right) on a curve fitting task.

**Sinc networks:** Our $\mathrm{sinc}$ networks all used a $\mathrm{sinc}$ activation in each layer following the literature (Ramasinghe et al., 2023; Saratchandran et al., 2024).

**ReLU Networks:** $\mathrm{ReLU}$ networks exhibit spectral bias (Rahaman et al., 2019). To overcome such a phenomenon positional embedding layers are often added to such a network (Tancik et al., 2020; Sitzmann et al., 2020; Saragadam et al., 2023). We followed the approach of those references and added a positional embedding layer to our $\mathrm{ReLU}$ network, which is a non-trainable layer that embeds the data into a higher dimensional space. It is well known this helps $\mathrm{ReLU}$ networks overcome spectral bias (Tancik et al., 2020). This also allowed us to consider a high dimensional problem as now the data embeds into a high dimensional space before it goes into trainable layers of the network.

**Curve fitting:** In the case of the curve fitting experiment, Sec. 6.1, we found that training for 200 epochs led to convergence. We trained all optimizers with a full batch of the data set.

**Image regression:** In the case of image regression, Sec. 6.2, we found that training for 5000 epochs led to convergence. We trained the optimizers SGD, Adam, and L-BFGS with a batch size of 256.

**Super image resolution:** For this experiment, Sec. 6.3, we found that training for 5000 epochs led to convergence. We trained the optimizers SGD, Adam, and L-BFGS with a batch size of 256.

**3D shape modelling:** For this experiment, Sec. 6.3, we found that training for 1000 epochs led to convergence. We trained the optimizers SGD, Adam, and L-BFGS with a batch size of 128 following Saragadam et al. (2023).

A.2.2 FURTHER EXPERIMENTS

**Iterations for curve fitting:** Fig. 11 shows the number of epochs each optimizer needed for convergence. As can be seen by that figure, as we add more width the number of epochs needed for each optimizer to converge went down suggesting it was easier for the optimizers to find global minima with more depth, though we did notice the gains went down as we added hidden layers past depth 4.

**Iterations for curve fitting:** Fig. 10 shows the number of iterations each optimizer needed for convergence. As can be seen by that figure, as we add more width the number of iterations needed for each optimizer to converge went down suggesting it was easier for the optimizers to find global minima with more width.

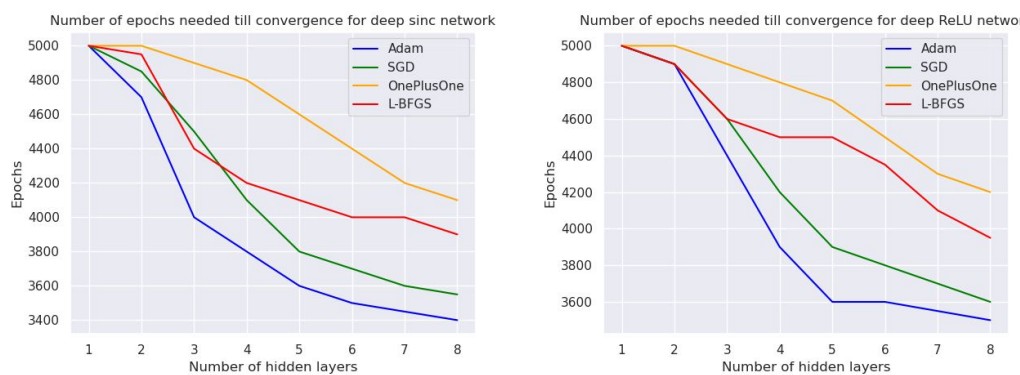

Figure 11: Number of epochs needed to converge for each optimizer as number of hidden layers increases for a deep sinc (left) and ReLU networks (right) on an image regression task.

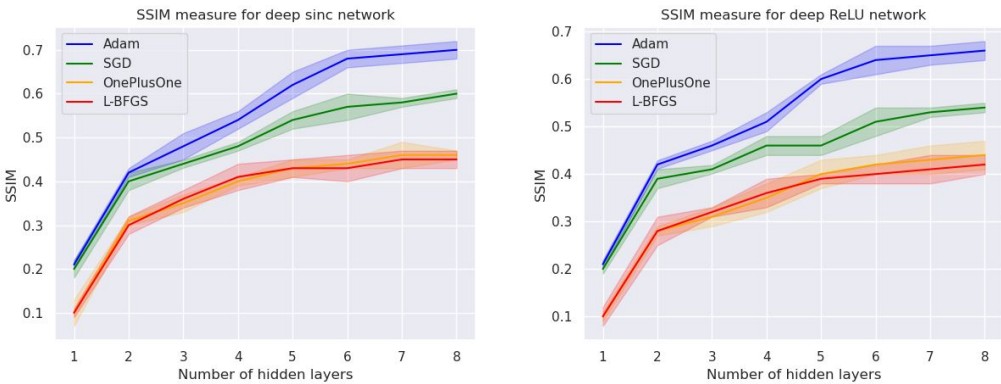

Figure 12: Final SSIM after convergence is plotted against the number of hidden layers for deep sinc (left) and ReLU (right) networks, each trained with four different optimizers on an image super resolution task. The results show that, for both network types, increasing the depth of the network consistently leads to higher test PSNR across all optimizers.

**Iterations for image regression:**

**Testing for image super resolution:** We computed the Structural Similarity Index Measure (SSIM) (Wang et al., 2004) after convergence to assess testing quality for the super image resolution task from Sec. 6.3. As shown in Fig. 12, increasing depth also enhanced the SSIM, but again, the gains plateaued after 4 hidden layers.

**Testing for Binary Occupancy fields:** We computed the Intersection Over Union (IOU) measure (Saragadam et al., 2023) for the binary occupancy experiment carried out in Sec. 6.4. Results can be seen in Fig. 13 shows the results showing that as more depth is added the IOU increases for all optimizers.

**Neural Radiance Fields (NeRF)** has recently gained attention as a powerful technique for modeling 3D scenes from multi-view 2D images using an MLP. NeRF operates by estimating the radiance field of a 3D scene given 3D coordinates $\mathbf{x} \in \mathbb{R}^3$ and viewing directions. The radiance field maps each input 3D point to its corresponding volume density $\sigma \in \mathbb{R}$ and directional emitted color $\mathbf{c} \in \mathbb{R}^3$.

Following the approach in the literature (Mildenhall et al., 2021; Xu et al., 2022; Chen et al., 2023), we trained NeRF models with both sinc and ReLU activations using the $\ell^2$ loss $\mathcal{L}_2$ defined in equation 5. In this experiment, the network depth was fixed at 8 layers, consistent with prior work, while

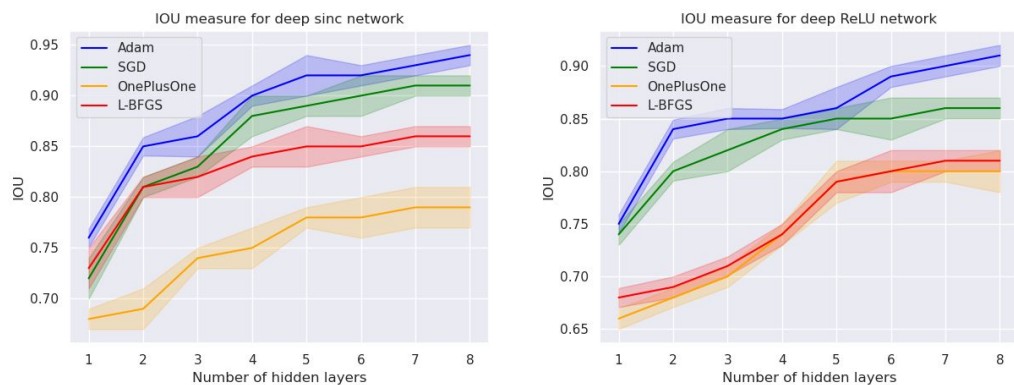

Figure 13: Final IOU after convergence is plotted against the number of hidden layers for deep sinc (left) and ReLU (right) networks, each trained with four different optimizers on a binary occupancy task. The results show that, for both network types, increasing the depth of the network consistently leads to higher test PSNR across all optimizers.

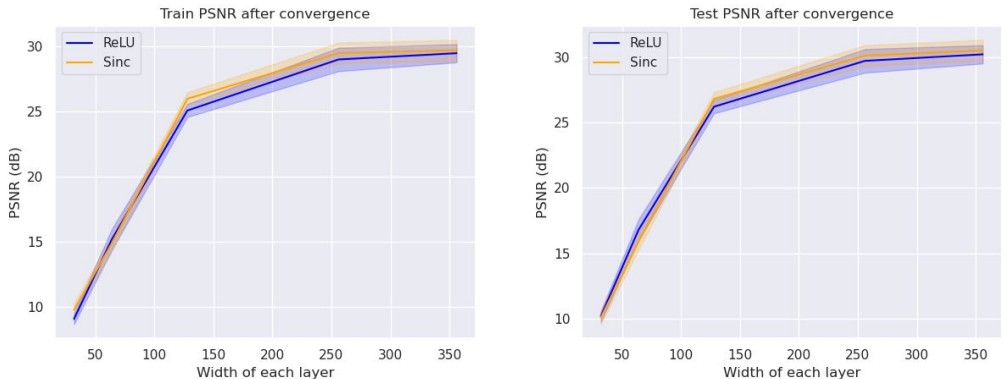

Figure 14: Final train (left) and test (right) PSNR after convergence for NeRF is plotted against the width of an 8 layer sinc and ReLU network, trained with Adam. The results show increasing the depth of the network consistently led to higher train PSNR and test PSNR.

we varied the width of each layer, testing sizes of 32, 64, 128, 256, and 356 neurons. During initial trials, we encountered difficulties training NeRF with SGD, OnePlusOne, and L-BFGS optimizers. After consulting the literature, we found that NeRF models are predominantly trained using Adam, and as Saratchandran et al. (2023) showed, training with L-BFGS is challenging due to issues with stochasticity.

Consequently, we employed Adam as the sole optimizer for this experiment. The training was conducted on the LLFF dataset from Mildenhall et al. (2021), which consists of eight instances, with three unseen views reserved for testing (Mildenhall et al., 2021). We calculated the PSNR by averaging across all eight training instances. For testing, we averaged the PSNR of each test view across the eight instances and then averaged over the three test views. As shown in Fig. 14, overparameterization consistently led to higher PSNR values in both the sinc and ReLU cases.

