# OpenReview forum: "Sampling Theory and Overparameterization: Shaping Loss Landscapes in $\ell^2$ Regression"
_ICLR.cc/2025/Conference — ICLR 2025 Conference Withdrawn Submission_

### Official Review · Reviewer_tSQh · 2024-10-21

**Soundness:** 3
**Presentation:** 4
**Contribution:** 3
**Rating:** 8
**Confidence:** 3

**Summary:**

This paper studies the advantages of overparameterization in feedforward neural networks from the perspective of sampling theory. By leveraging this approach, the authors derive general properties of the loss landscape that hold regardless of the optimization algorithm used. They demonstrate that overparameterization results in an exponential increase in the number of global minima near the origin and an expansion of the dimensionality of the loss valley, leading to "wider" global minima in some sense. The authors also conducted experiments on 4 classical regression tasks to illustrate their point

**Strengths:**

The authors present an insightful and, to the best of my knowledge, novel connection to the Nyquist-Shannon-Whittaker sampling theory, which they use to derive general properties of the loss landscape independently of the optimization algorithm. This approach differs from the more conventional methods that focus on specific variations of gradient descent, potentially opening up a new research avenue for even more advanced findings in the future.
The paper is well-written and easy to follow, with clear proof sketches that provide an intuitive understanding of the key concepts. Overall, I found the paper to be both compelling and well-structured.

**Weaknesses:**

The experiments are valuable in illustrating that the benefits of over-parameterization are not tied to a specific optimization method. However, they feel somewhat disconnected from the theoretical results and highlight a gap in the theoretical findings. Theorems 5.2 and 5.3 focus on the number of global minima, while the experiments assess the quality of the found minima in terms of generalization. Based on the theorems, as mentioned in line 372, over-parameterization should primarily help optimizers converge more efficiently, which aligns with the experiment on the number of epochs, relegated to the appendix.
The only theoretical result that might suggest a link between over-parameterization and better generalization is Theorem 5.5, which states that the dimensionality of the valleys grows linearly with the number of parameters. This might relate to the observation that wider minima tend to generalize better, but this connection is not explicitly made in the paper. It would be helpful to clarify this, especially since one could argue that the "width" of the valley could be considered in relation to the ratio of the dimensionality of the valley to the dimensionality of the parameters, both of which are growing linearly. The paper would benefit significantly from a direct theoretical result showing how over-parameterization improves generalization.

**Questions:**

Main clarifications and suggestions:
- Theorem 5.3: I am slightly surprised here by the uniform convergence result since only continuity is assumed on the squared integrable target function $f$. To my knowledge, typical Weirstrass-like theorems require stronger assumptions for uniform convergence. I briefly checked the appendix, and line 909 seemed a bit unclear. It states: "Then using the fact that L2-convergence implies pointwise convergence (Stein & Shakarchi, 2009), we have that for any x ∈ [0, 1] it holds |f − g| <ϵ/4". As stated, the theorem discusses pointwise convergence, but you conclude with uniform convergence. I haven't looked too much in detail, though, and I might be missing something. Could you clarify?
- Definition 5.1: I feel like some concept is missing in the definition of the global minimum valley, maybe something along the lines of connexity. According to the current version, the union of two potentially separate valleys is still a valley, which is counter-intuitive and also may raise some technical difficulties in defining what is referred to afterward as "distinct" valleys (which should be properly defined as well)

Minor
- In section 4, you could maybe add a formula to define the feedforward neural network $\mathcal{N}$.
- Theorem 5.3: why do you restrict x to be different from the $x_i$ ?
- Experiments: Could you define the PSNR metric?

---

### Official Review · Reviewer_wrhg · 2024-10-23

**Soundness:** 1
**Presentation:** 1
**Contribution:** 2
**Rating:** 3
**Confidence:** 2

**Summary:**

This paper studies influence of overparameterization on the loss landscape of $l^2$ supervised regression problems, independent of any specific optimizer. Authors claim that overparameterization not only exponentially increases the number of global min valleys but also expands the dimensionality of loss valleys for feedforward neural networks with ReLU and sinc activation.

**Strengths:**

At first glance the contribution seems interesting but the presentation of the paper and main results are very confusing.

**Weaknesses:**

There are many confusing points in the paper that makes it difficult to follow the paper.
I think before clarifying on the mentioned questions below, it is not possible to evaluate this work confidently.

**Questions:**

1-In Theorem 5.2 it is stated that `Consider N (x; θ) to be a shallow neural network with n neurons in its hidden layer.'
It is not clear for me what n stands for? Wasn't it the number of training examples? But why authors mention it is number of hidden neurons? Also for the sentence 'The dependence on n reflects the number of neurons .... '. (same for Theorem 5.3) Also, it is not specified what is the dimension of input, output, network size, ... ?

2-Proof overview line 252: Could you please elaborate how this assumption makes sense while this assumption has been never mentioned  before their Theorem statement?

3-Throem 5.2 (Line 242): How can authors make sure that there exists any global minimum valleys of L-2 at first glans?  There might be just isolated global minima?

4-In Section 6, authors claim that their results in Figures 3 and 4 are supporting their theories.
Could you please elaborate how: For example better performance of networks with larger size might come because larger networks have
higher approximation power to estimate the target functions. It is not clear if really the better performance is coming from optimization aspects.

5- Lines 189-194: By increasing the parameters of a network, the search function  space also increase exponentially. So, why having more global min can guarantee better performance? Note that our search space is also expanded too much? Could you please elaborate on this this?

6-PSNR never defined?

7-Strange notations are used for example for the network.

---

### Official Review · Reviewer_Usa2 · 2024-10-27

**Soundness:** 1
**Presentation:** 2
**Contribution:** 1
**Rating:** 1
**Confidence:** 3

**Summary:**

This paper studies the l2 training loss landscape of shallow sinc-activated neural networks for interpolation of 1-D signals. The motivation stems from the classical Nyquist-Shannon-Whittaker (NSW) sampling theorem which asserts that uniformly spaced samples are sufficient for the lossless reconstruction of continuous band-limited signals. NSW is constructive in that it asserts that the reconstruction can be done via a combination of appropriately scaled and shifted sinc functions. The authors consider the setting where the scalings and shifts of these sinc functions are made learnable.

The two main results of the paper state that
1. the number of global minima in the training loss landscape increases exponentially in the number of hidden neurons.
2. the dimensionality of global minima of ReLU networks grows in the width and depth of the network.

Their high level claim is that these two results indicate that the generalization of neural networks benefits from overparameterization.

**Strengths:**

The paper is for the most part decently written, though there are some missing references (see below).

The high-level idea of taking classical closed-form schemes and untying the parameters to make them learnable is sound.
From my perspective, it might very well be possible that such sinc networks can sometimes outperform NSW in the aliasing (e.g. finite sample) regime (even though the authors make no such claim/motivation).

**Weaknesses:**

- Lack of references:

There is a lack of references concerning existing studies of the loss-landscape of neural networks. For example, similarities and differences to [1] should be discussed, since that paper seems to study a very similar question.

- Poor motivation:

It is unclear whether untying the parameters in NSW can actually benefit generalization. At the very least, there should be some experiments that compare vanilla NSW reconstruction with trained sinc networks, showing that the latter can achieve lower test error (e.g. for some simple class of low-pass signals). In fact, the authors never properly define the test error, which in this setting should be the integral of the squared error over some continuous time interval.

There are not many papers in the literature that study sinc networks. It is worth noting that both of the two references in the paper (by Ravanelli and Bengio) are workshop papers. I suspect that networks with such activations can be hard to train in practise. This point should at least briefly be discussed since efficient optimization is an important aspect for the success of machine learning algorithms.

- Severe misunderstanding of the implications of the geometry of the loss landscape for generalization:

An abundance of global minima is neither sufficient, nor even indicative of good generalization. To reach such conclusions, one must, in my view, either prove—or at least provide some empirical evidence— that either
1. a vast majority of global minima admits good generalization (and then “hope” that the optimizer selects “typical” global minima).
2. the optimizer selects only well generalizing global minima, independent of the number or characteristics of global minima.

Neither of the above things are taken into consideration. As highlighted, e.g., in line 60, the authors consider an abundance of global training minima as beneficial in its own right:

> “What is particularly interesting about this result is that it is independent of any optimizer, implying that for such networks, overparameterization provides a significant benefit for the loss landscape that should help any optimizer.”

However, this is generally not true if one simply enriches the landscape with many poorly generalizing global minima, as this will most likely make it harder for an optimizer to find well-generalizing solutions. A simple example is learning linear functions with empirical risk minimization over polynomials. One can trivially generate more interpolating solutions (global minima) by increasing the maximum degree of polynomials— however, almost all of these interpolators will generalize poorly.

In a similar manner, in Theorem 5.2, the number of interpolating solutions is increased by adding sinc functions centered outside the window [0,1] such that they evaluate to zero at the training sample. Notably, these sincs are generally non-zero over [0,1] outside the training points. Intuitively, most choices of such extraneous sinc functions will hurt generalization, since their side-lobes will leak into the interval [0,1]. The authors provide no insight into how harmful leakage might be avoided, neither explicitly nor implicitly by showing good generalization via experiments.

Theorem 5.5 suffers from essentially the same technical issue in that it artificially increases the number of global minima (almost all of which generalize poorly) by constructing indicator functions (with triplets of neurons), which are supported outside the training set.

- Misunderstanding of the connection between overparametrization and overfitting:

The authors state in line 105 that
> “While overfitting was once a concern, studies like Zhang et al. (2021) showed that overparameterized networks can still generalize well, despite perfectly fitting training data”.

Overfitting is still very much a concern in general learning settings with neural networks! It is not the case that neural networks automatically avoid overfitting in all scenarios (i.e. for all distributions and optimizers). This follows for example from the no-free lunch theorem of learning, or, empirically, from the fact that neural net classifiers can interpolate random labels. Zhang et al showed empirically that some ReLU networks trained with SGD tend to avoid overfitting on some standard CV datasets. This does certainly not imply that sinc-activated networks trainined on band-limited signals admit a favourable implicit bias that prevents overfitting.

- Minor comments:

Throughout: “< >” -> \langle \rangle

----------------------------------------------------------------
[1]: Simsek, Berfin, et al. "Geometry of the loss landscape in overparameterized neural networks: Symmetries and invariances." International Conference on Machine Learning. PMLR, 2021.

**Questions:**

- Experiments (these should all be very easy to implement!):
    - Can a trained sinc network empirically even beat vanilla NSW reconstruction?
    - Can SGD actually find global training minima of sinc networks?
    - I suggest including figures that show, separately, the training error, the actual test error on unseen points on the continuous interval [0,1] (e.g. sampled uniformly at random from [0,1]) and, possibly, the “test” error on unseen points on the uniform grind (as in fig. 3)
    - Why is the “test” error on unseen points on the uniform grid (as considered in Fig. 3) even relevant? Perhaps I am missing something.
- Do you have an insight into how an optimizer (be it a general one, or e.g. gradient descent) could avoid selecting bad global minima in your setting?
- How do your kind of results compare to the ones of [1]?

---

### Official Review · Reviewer_ciLo · 2024-10-31

**Soundness:** 2
**Presentation:** 3
**Contribution:** 2
**Rating:** 5
**Confidence:** 4

**Summary:**

This paper investigates the effect of overparametrization of neural networks on the loss landscape for supervised l2 regression, and tries to provide an explanation for why overparameterization is helpful for loss minimization via Nyquist-Shannon-Whittaker sampling theory. Unlike much of existing literature, this paper does not focus on gradient-based optimization but rather explores how overparameterization reshapes the loss landscape independently of the optimization algorithm used. The authors essentially claim that overparameterization of deep neural networks (i.e. as they theoretically demonstrate in the case of adding neurons to the hidden layer of a shallow network and in separate theorems, adding more layers to a given network) tends to increase the number of global minima of the loss landscape.

**Strengths:**

Strengths:
- The paper introduces a new theoretical approach to explain how overparametrization in neural networks help with its training. They do so by drawing connections to NSW sampling theorem and analyze sinc and Relu activated networks in a noble attempt.

- The framework is analyzed independently of a specific optimizer choice (using first order gradients, second order gradients or gradient free), which gives a broad perspective.

**Weaknesses:**

- Whereas the paper attempts to provide an analysis for how the global minima of loss landscapes change with additional neurons, I find their arguments not convincing enough and ill-justified. Primarily, the reason
for nonconvex optimization problems in neural network training being hard is not because of the non-existence of sufficient global minima, but usually because the optimizers usually get stuck in some local minima. Moreover, the plots in the paper demonstrate that as the models get bigger, the PSNR tends to increase. However, this does not verify the theoretical claims in the paper, as these improvements can simply be due to the fact that the networks become more expressive and are better at handling data.  I find the connection of having more global minima lacking in these experiments. In addition, as the neuron numbers are increased either in depth or width, the model parameter count will also increase. Therefore, the loss landscape will be in a different dimension. Hence, is it a fair comparison to say that in the higher dimensional parameter space the loss landscape will have more global minima now?

- Moreover, in practical settings, why does having more global minima help, and do different global minima have the same generalization property? The generalization properties of the newly introduced global minima need further elaboration. Furthermore, the paper lacks the analysis of drawbacks of overparameterization during training, such as increased time and memory constraints to train the neural networks.

In summary:

- The paper does not analyze the generalizaton properties of the new global minima, and whether having more global minima always corresponds to having lower test error is unclear.

- The experimental results do not directly verify the theoretical claims. How do you show that the PSNR gets better as model size increases because of the increased number of global minima, i.e. are global minima even relevant in those neural network experiments?

- The paper fails to consider drawbacks of overparametrization, such as increased compute and memory cost.

**Questions:**

- It would be good to address the main comment re. this paper that the claims are not sufficiently validated through either theory or experiments for the reasons cited above, due to an "apples to oranges" comparison when one adds more neurons in width or depth to the baseline case.  It would be good to just give "absolute" results in terms of what happens to the loss landscape as a function of the number of neurons in the network, rather than provide the "differential" results as comparisons to a baseline case, as the two settings are not the same.

- Can you clarify the connection between the trends seen in the experiments with your theoretical results? Why do you attribute the increase in PSNR as model size grows to an increased number of global minima? Can’t the model be settled in another local minima that’s just better in those cases?

- Does this approach work only for MLPs? Can you not train superresolution instances with CNNs as well? It might be more helpful and realistic for image superresolution.

- How do the generalization properties of the new global minima affect the test performance upon training an overparameterized neural network (say you add l neurons to a shallow sinc neural network or 3l neurons to a Relu neural network)?

- Can you share the experimental figures (results) for your experiments (other than your PSNR curves)?

Minor comments:
- On page 29, PNSR → PSNR

- Throughout the paper, both “data set” and “dataset” phrases are used. While both are grammatically acceptable in ML papers, it is good to be consistent and stick to one.

---

### Official Review · Reviewer_L8hh · 2024-11-04

**Soundness:** 3
**Presentation:** 3
**Contribution:** 3
**Rating:** 6
**Confidence:** 3

**Summary:**

This paper investigates the influence of overparameterization in neural networks on the loss landscape of $l_2$ supervised regression problems, without assuming any specific optimization algorithm. By leveraging the Nyquist-Shannon-Whittaker sampling theorem, the authors establish a theoretical connection between sampling theory and overparameterized neural networks, revealing that overparameterization exponentially increases the number of global minima and expands loss valleys in feedforward networks. Empirical results validate these theoretical insights using multiple optimization methods, offering new perspectives on the benefits of overparameterization beyond gradient descent.

**Strengths:**

1. The paper provides a new perspective by connecting the overparametrization and with NSW sampling theory.

2. The paper provides theoretical results about how the number of distinct global minimum valleys grow exponentially with respect to the increasing number of neurons in a single layer in a shallow neural network or the increasing number of layers in a deep neural network. The proofs are very constructive and novel.

**Weaknesses:**

1. The ending statement ``grows at least exponentially in $l$" in the main result Theorem 5.2 is a bit hand-waving. Can it be more quantitative? I look at the cited paper by Chamizo and he provided precise bounds.

2. The provided experiments have unclear relevance to the theoretical findings. A relevant experiment to consider would be examining how the number of global minimum valleys scales with respect to the width and/or number of hidden layers. Even with a simplified model, a verification experiment of this kind would help substantiate the theoretical predictions.

**Questions:**

1. What does $p$ refer to in eqn. (5)?

2. In Section 6, what is PSNR? Please explain the motivation to consider PSNR in your experiments.

3. I suggest reorganizing the proofs in the appendix to align with the steps outlined in the main text. Referring each step in the main text to its corresponding proof in the appendix would make it easier for readers to follow the argument and verify each result.

4. Does the generalization result in Theorem 5.3 work for deep feedforward neural networks?

---

### Note · Authors · 2024-11-30

I have read and agree with the venue's withdrawal policy on behalf of myself and my co-authors.